# ViPO: Visual Preference Optimization at Scale

**Ming Li**[1,2,‡]**, Jie Wu**[2]**, Justin Cui**[2,3]**, Xiaojie Li**[2]**, Rui Wang**[2]**, Chen Chen**[1]
[1]University of Central Florida, [2]ByteDance Seed, [3]UCLA

## Abstract

While preference optimization is crucial for improving visual generative models, how to effectively scale this paradigm for visual generation remains largely unexplored. Current open-source preference datasets typically contain substantial conflicting preference patterns, where winners excel in some dimensions but underperform in others. Naively optimizing on such noisy datasets fails to learn meaningful preferences, fundamentally hindering effective scaling. To enhance the robustness of preference algorithms against noise, we propose Poly-DPO, which extends the DPO objective with an additional polynomial term that dynamically adjusts model confidence during training based on dataset characteristics, enabling effective learning across diverse data distributions from noisy to trivially simple patterns. Beyond biased patterns, existing datasets suffer from low resolution, limited prompt diversity, and imbalanced distributions. To facilitate large-scale visual preference optimization by tackling key data bottlenecks, we construct ViPO, a massive-scale preference dataset with 1M image pairs (1024px) across five categories and 300K video pairs (720p+) across three categories. Leveraging state-of-the-art generative models and diverse prompts ensures consistent, reliable preference signals with balanced distributions. Remarkably, when applying Poly-DPO to our high-quality dataset, the optimal configuration converges to standard DPO. This convergence validates both our dataset quality and Poly-DPO's adaptive nature: sophisticated optimization becomes unnecessary with sufficient data quality, yet remains valuable for imperfect datasets. We comprehensively validate our approach across various visual generation models. On noisy datasets like Pick-a-Pic V2, Poly-DPO achieves 6.87 and 2.32 gains over Diffusion-DPO on GenEval for SD1.5 and SDXL, respectively. For our high-quality ViPO dataset, models achieve performance far exceeding those trained on existing open-source preference datasets. These results confirm that addressing both algorithmic adaptability and data quality is essential for scaling visual preference optimization. Code, models and open-source datasets will be released at: *https://github.com/liming-ai/ViPO*.

## 1 Introduction

Preference optimization techniques, such as Reinforcement Learning from Human Feedback (RLHF) Ouyang et al. (2022) and Direct Preference Optimization (DPO) Rafailov et al. (2023), have proven essential for aligning large-scale models with human values. Building on this success in language models, researchers have extended these paradigms to visual generation. Among various approaches, off-policy methods like Diffusion-DPO Wallace et al. (2024) are particularly promising for large-scale applications. Unlike on-policy RL approaches Xu et al. (2023); Liang et al. (2025); Liu et al. (2025a); Xue et al. (2025); Black et al. (2024) that require costly iterative sampling, off-policy methods leverage pre-collected preference datasets without expensive policy deployment, making them inherently more suitable for scaling Wu et al. (2025a). However, while preference optimization is crucial for improving visual generative models, how to effectively scale this paradigm remains largely unexplored.

We argue that the primary obstacle to scaling lies in the conflicting preference patterns prevalent in current datasets. Specifically, existing open-source preference datasets Wu et al. (2023b;a); Ma et al. (2025); Kirstain et al. (2023) are usually constructed by early diffusion models, contain substantial conflicts where winner images excel in certain dimensions (e.g., aesthetics) but underperform in others (e.g., text-image alignment). Naively optimizing on such noisy datasets fails to learn meaningful preference patterns, fundamentally hindering effective scaling of preference optimization. Without proper handling of these conflicting signals, models struggle to extract genuine preference pattern, leading to suboptimal

---

‡ This work was done during the first author's internship at USA

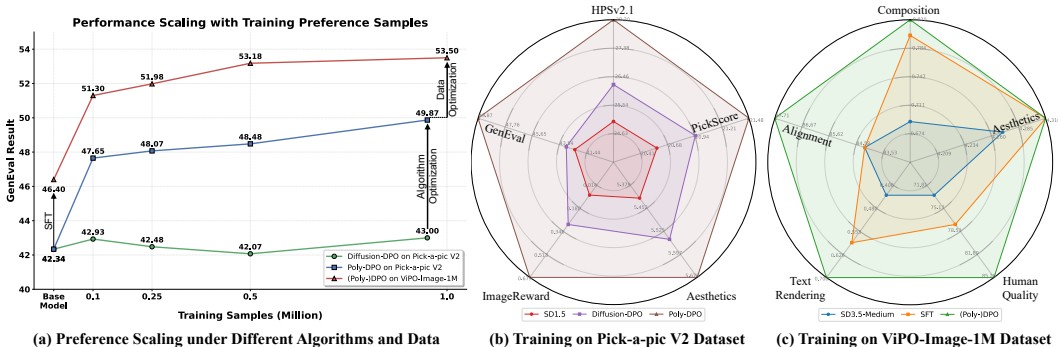

Figure 1: (a) Preference scaling with our Poly-DPO and ViPO-Image-1M dataset. (b) When training on a biased preference dataset such as Pick-a-pic V2, our Poly-DPO outperforms Diffusion-DPO in all evaluation dimensions. (c) Our proposed ViPO-Image-1M dataset can comprehensively improve the SD3.5-Medium.

performance that fails to further improve with data scale, as demonstrated in Figure 1 (a). Beyond biased preference patterns (conflict or over-simple samples), existing datasets suffer from multiple limitations: low visual resolution (typically 512-768), limited prompt diversity, imbalanced data distributions from random collection strategy, and constraints from outdated generation models, as shown in Table 1. These factors collectively hinder the effective scaling of preference learning.

To better learn from biased preference datasets, we propose Poly-DPO, which extends Diffusion-DPO with a polynomial term that dynamically adjusts sample weighting based on prediction confidence. This mechanism enables effective learning across diverse data characteristics: for existing datasets that contain conflicting preferences (e.g., Pick-a-pic V2), it helps models focus on informative samples despite contradictory signals and improves the final generation quality as shown in Figure 1 (b). To comprehensively address data quality barriers, we construct ViPO, a massive-scale and high-quality visual preference dataset comprising 1M image pairs (1024px) across five categories and 300K video pairs (720p+) across three categories. By leveraging state-of-the-art generative models (FLUX Labs (2024), Qwen-Image Wu et al. (2025a), WanVideo Wan et al. (2025)) and systematic categorization, we ensure reliable, balanced preference signals that enable robust preference learning at scale.

Extensive experiments validate the synergy between our contributions. On noisy datasets like Pick-a-Pic V2, Poly-DPO significantly outperforms standard Diffusion-DPO by handling conflicting preference patterns. Training on our ViPO dataset, the SD1.5 model achieves state-of-the-art results far exceeding those trained on existing datasets in Figure 1 (a) and comprehensively improves the SD3.5-Medium as shown in Figure 1 (c). Remarkably, when applied to ViPO-Image-1M, Poly-DPO converges to standard DPO ($\alpha \to 0$) and remains robust across a neighborhood around zero, indicating it works equally well on high-quality data without tuning. This convergence mutually validates both contributions: ViPO's quality enables stable optimization across different $\alpha$ values, while Poly-DPO adaptively simplifies through a single hyperparameter when data quality permits. These findings show that scaling visual preference optimization requires both algorithmic robustness for imperfect data and systematic data curation.

Our contributions are summarized as follows:

- *New Insight for Visual Preference Scaling*: We demonstrate that the biased preference distributions characterized by conflicting patterns constitute a fundamental bottleneck for preference scaling. We reveal that standard Diffusion-DPO fails to extract effective signals from such data, leading to performance saturation despite data scaling.

- *Poly-DPO Optimization Algorithm*: We introduce Poly-DPO, which dynamically adjusts sample weighting based on confidence levels, enabling effective learning from conflicting patterns in noisy datasets while preventing over-confidence on trivially distinguishable preferences.

- *Large-Scale High-Quality Dataset*: We construct ViPO dataset with 1M high-resolution image pairs and 300K video pairs using state-of-the-art models and systematic categorization, providing reliable and balanced preference signals that establish a new benchmark for preference learning at scale.

- *Mutual Validation of Approach*: Our experiments demonstrate that Poly-DPO excels on biased datasets while converging to standard DPO ($\alpha \to 0$) with robustness across neighboring $\alpha$ values on high-quality ViPO-Image-1M data, confirming that sophisticated optimization becomes unnecessary with sufficient data quality yet remains essential for imperfect datasets.

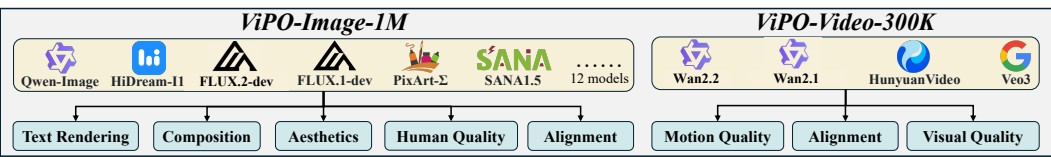

Figure 2: Overview of our ViPO-Image-1M and ViPO-Video-300K dataset.

| Dataset | Prompt | Image/Video | Pair | Resolution | Construction | Generative Models |
|---|---|---|---|---|---|---|
| *Image Dataset* | | | | | | |
| HPDv1 | 25,205 | 98,807 | 25,205 | [512-960] | Random | SD1.4 |
| HPDv2 | 103,700 | 430,060 | 645,090 | [480-640] | Random | SD2.0, CogView2, DALL-E 2 |
| Pick-a-pic v1 | 37,523 | 623,694 | 583,747 | [512-768] | Random | SD2.1, SDXL, Dreamlike, etc |
| Pick-a-pic v2 | 58,960 | 928,068 | 959,040 | [512-768] | Random | SD2.1, SDXL, Dreamlike, etc |
| HPDv3 | 202,274 | 1,088,274 | **1.17M** | [256-1024] | Random | SD1.4, SDXL, FLUX.1 dev, etc |
| **Ours** | **1,000,000** | **2,000,000** | 1.00M | **1024** | **Categoried** | **Qwen-Image, HiDream-I1, etc** |
| *Video Dataset* | | | | | | |
| VideoDPO | 10,000 | 20,000 | 10,000 | 480p | Random | CogVideo, VideoCrafter2, etc |
| **Ours** | **30,000** | **60,000** | **30,000** | **720p, 1024p** | **Categoried** | **WanVideo, Veo3, etc** |

Table 1: Comparison with existing open-source preference datasets.

## 2 RELATED WORKS

**Diffusion-based Visual Generation.** Building upon pioneering diffusion models Sohl-Dickstein et al. (2015); Ho et al. (2020); Song et al. (2021a;b); Lipman et al. (2023) and their successful scaling Rombach et al. (2022); Ho & Salimans (2022); Dhariwal & Nichol (2021), visual generation has achieved remarkable progress. Advanced models like FLUX Labs (2024), Qwen-Image Wu et al. (2025a) for images, and HunyuanVideo Kong et al. (2024), WanVideo Wan et al. (2025) for videos, have enabled stunning visual content creation across diverse applications Zhang et al. (2023); Mou et al. (2024); Ruiz et al. (2023); Ye et al. (2023); Brooks et al. (2023); Li et al. (2025). Despite these advances, two key challenges remain: aligning outputs with complex user prompts and optimizing multiple quality dimensions simultaneously.

**Reinforcement Learning from Human Feedback (RLHF).** RLHF has demonstrated remarkable success in aligning large language models with human values Ouyang et al. (2022); Touvron et al. (2023a); Bai et al. (2023); Wang et al. (2024); Team et al. (2025). Current approaches fall into two categories: on-policy methods (PPO Schulman et al. (2017), GRPO Shao et al. (2024)) that require iterative sampling and reward model evaluation during training, and off-policy methods (DPO Rafailov et al. (2023)) that learn directly from pre-collected preference datasets. Off-policy methods avoid the computational overhead of online sampling, making them more efficient Touvron et al. (2023b); Rafailov et al. (2023), though their effectiveness depends on preference dataset quality Morimura et al. (2024); Wu et al. (2025a).

**Reinforcement Learning for Visual Generation.** Recent research extends RL success from LLMs to visual generation. On-policy methods include ReFL-based approaches Xu et al. (2023); Clark et al. (2024); Li et al. (2024a); Wu et al. (2025b) that integrate reward maximization into diffusion training, and PPO-based methods Black et al. (2024); Xue et al. (2025); Liu et al. (2025a) that model diffusion as an MDP. However, these face scalability constraints from computational intensity and reward hacking vulnerability. Off-policy methods, particularly DPO-based approaches Wallace et al. (2024); Yang et al. (2024); Dong et al. (2024); Liu et al. (2025b); Karthik et al. (2025); Zhu et al. (2025); Zhang et al. (2025a); Liu et al. (2026), offer superior computational scalability by training on preference pairs without online sampling, but they require high-quality preference datasets and effective optimization algorithms.

## 3 DIFFUSION PREFERENCE OPTIMIZATION WITH POLY-DPO

### 3.1 PRELIMINARIES FOR DIFFUSION-DPO

**Diffusion Models.** Denoising diffusion models operate through two complementary processes: a forward process that progressively corrupts data by introducing noise, and a reverse process that reconstructs clean data from the corrupted versions. Specifically, during the forward process, a clean data point $\mathbf{x}$ undergoes noise corruption at timestep $t \in [0,1]$, resulting in a conditional distribution $q(\mathbf{x}_t|\mathbf{x})$ characterized by $\mathbf{x}_t = \alpha_t \mathbf{x} + \sigma_t \boldsymbol{\epsilon}$, where $\boldsymbol{\epsilon} \sim \mathcal{N}(\mathbf{0}, \mathbf{I})$, $\alpha_t, \sigma_t$ represent predefined noise scheduling parameters, and $\lambda_t = \log(\alpha_t^2/\sigma_t^2)$ denotes the logarithmic signal-to-noise ratio (SNR). With the input condition $c$, the training process optimizes a weighted noise prediction objective formulated as:

$$\mathcal{L}_{\text{DM}}(\mathbf{x}) = \mathbb{E}_{t \sim \mathcal{U}(0,1), \boldsymbol{\epsilon}} \left[ -w_t \lambda_t' ||\boldsymbol{\epsilon}_\theta(\mathbf{x}_t; c, t) - \boldsymbol{\epsilon}||_2^2 \right], \qquad (1)$$

where $w_t$ represents a weighting function and $\lambda_t' = d\lambda/dt$. Notably, most diffusion and flow matching training objectives can be expressed in the form of Eq. (1) through appropriate choices of $w_t$ and $\lambda_t$.

**Reward Models.** For a given image $\mathbf{x}$ and input conditioning $\mathbf{c}$, a reward model $R(\mathbf{x},\mathbf{c})$ represents a function that quantifies the quality of the generated output. A widely adopted framework for modeling human preferences is the Bradley-Terry (BT) model, which establishes the preference probability distribution over a triplet $(\mathbf{c},\mathbf{x}^w,\mathbf{x}^l)$: $P(\mathbf{x}^w \succ \mathbf{x}^l|\mathbf{c}) := \sigma\left(R(\mathbf{x}^w,\mathbf{c}) - R(\mathbf{x}^l,\mathbf{c})\right)$, where $\sigma$ denotes the sigmoid function, and $\mathbf{x}^w$, $\mathbf{x}^l$ represent the winner and loser images, respectively. The objective of reward fine-tuning is to optimize the diffusion model $p_\theta$ such that it maximizes the expected reward of generated outputs while incorporating KL regularization $D_{\text{KL}}$ to prevent reward over-optimization: $\max_\theta \ \mathbb{E}_{\mathbf{c},\mathbf{x}\sim p_\theta(\mathbf{x}|\mathbf{c})}\left[R(\mathbf{x},\mathbf{c})\right] - \beta D_{\text{KL}}\left(p_\theta(\cdot|\mathbf{c})\|p_{\text{ref}}(\cdot|\mathbf{c})\right)$ where $p_{\text{ref}}$ is a reference model and $\beta$ is a hyperparameter that controls the strength of KL regularization.

**Diffusion-DPO.** Following the DPO framework Rafailov et al. (2023), the training objective can be reformulated to enable direct optimization through the conditional distribution $p_\theta(\mathbf{x}|\mathbf{c})$:

$$L_{\text{DPO}}(\theta) = -\mathbb{E}_{(\boldsymbol{x}^w,\boldsymbol{x}^l)}\left[\log \sigma\left(\beta \log\frac{p_\theta(\boldsymbol{x}^w)}{p_{\text{ref}}(\boldsymbol{x}^w)} - \beta \log\frac{p_\theta(\boldsymbol{x}^l)}{p_{\text{ref}}(\boldsymbol{x}^l)}\right)\right]. \tag{2}$$

However, directly applying Eq. (2) to diffusion models presents a fundamental challenge, as the log-likelihoods of diffusion models are intractable. To address this limitation, Diffusion-DPO Wallace et al. (2024) introduces an approximation that connects the diffusion denoising process with the forward training objective in Eq. (1). Specifically, at timestep $t$, the log-likelihood ratio can be approximated as:

$$\log \frac{p_\theta(\boldsymbol{x})}{p_{\text{ref}}(\boldsymbol{x})} \approx -w_t\lambda_t'\left(\|\boldsymbol{\epsilon}_\theta(\mathbf{x}_t;\mathbf{c},t) - \boldsymbol{\epsilon}_t\|_2^2 - \|\boldsymbol{\epsilon}_{\text{ref}}(\mathbf{x}_t;\mathbf{c},t) - \boldsymbol{\epsilon}_t\|_2^2\right). \tag{3}$$

By substituting Eq. (3) into Eq. (2), we obtain the final Diffusion-DPO loss function:

$$\begin{aligned}L_{\text{Diffusion-DPO}}(\theta) = -\mathbb{E}_{(\boldsymbol{x}^w,\boldsymbol{x}^l),\boldsymbol{\epsilon}_t,t}\Big[\log\sigma\Big(&-\beta w_t\lambda_t'\big((\|\boldsymbol{\epsilon}_\theta(\mathbf{x}_t^w;\mathbf{c},t) - \boldsymbol{\epsilon}_t\|_2^2 - \|\boldsymbol{\epsilon}_{\text{ref}}(\mathbf{x}_t^w;\mathbf{c},t) - \boldsymbol{\epsilon}_t\|_2^2)\\&-(\|\boldsymbol{\epsilon}_\theta(\mathbf{x}_t^l;\mathbf{c},t) - \boldsymbol{\epsilon}_t\|_2^2 - \|\boldsymbol{\epsilon}_{\text{ref}}(\mathbf{x}_t^l;\mathbf{c},t) - \boldsymbol{\epsilon}_t\|_2^2)\big)\Big)\Big].\end{aligned} \tag{4}$$

### 3.2 Poly-DPO: Polynomial Expansion for Preference Optimization

**Diffusion-DPO as the Binary Classification Task.** Building upon the Diffusion-DPO framework, we propose Poly-DPO, which leverages insights from poly loss Leng et al. (2022) design to enhance preference learning. We begin by reinterpreting the standard Diffusion-DPO objective into the standard binary classification task. Specifically, we can define the preference probability:

$$p^{w>l} = \sigma\left(\beta \log\frac{p_\theta(\boldsymbol{x}^w)}{p_{\text{ref}}(\boldsymbol{x}^w)} - \beta \log\frac{p_\theta(\boldsymbol{x}^l)}{p_{\text{ref}}(\boldsymbol{x}^l)}\right), \tag{5}$$

which quantifies the model's relative preference for the winner image $\boldsymbol{x}^w$ over the loser image $\boldsymbol{x}^l$ compared to the reference model. This allows us to rewrite the Diffusion-DPO loss as:

$$L_{\text{Diffusion-DPO}}(\theta) = -\mathbb{E}_{(\boldsymbol{x}^w,\boldsymbol{x}^l)\sim\mathcal{D}}\left[\log(p^{w>l})\right]. \tag{6}$$

This reformulation reveals that Diffusion-DPO can be regarded a cross-entropy loss for binary classification, where the model learns to maximize the probability of correctly ranking preferred generations.

**Polynomial Expansion of Preference Learning.** Inspired by poly loss Leng et al. (2022), we can get the Taylor expansion of the standard cross-entropy loss in the context of Diffusion-DPO framework:

$$L_{\text{Diffusion-DPO}}(\theta) = -\log(p^{w>l}) = \sum_{j}^{\infty}\frac{1}{j}(1-p^{w>l})^j = 1\times(1-p^{w>l})^1 + \frac{1}{2}\times(1-p^{w>l})^2 \ ... \tag{7}$$

The core idea of the Poly Loss is to add a perturb term $\alpha_j$ for the Top-N polynomials that contribute the most to the gradient and keep others, and we can obtain the Poly-N loss:

$$\begin{aligned}L_{\text{Poly-N}} &= \underbrace{(1+\alpha_1)(1-p^{w>l})^1 + ... + (1+\alpha_N/N)(1-p^{w>l})^N}_{\text{perturbed by }\alpha_j} + \underbrace{1/(N+1)(1-p^{w>l})^{N+1} + ...}_{\text{same as }L_{\text{CE}}}\\&= -\log(p^{w>l}) + \sum_{j}^{N}\alpha_j(1-p^{w>l})^j.\end{aligned} \tag{8}$$

| 1 hyper-parameter | 2 lines of code | 3 preference distributions |
|---|---|---|
| $L_{\text{Diffusion-DPO}} = -\log\left(p^{w>l}\right)$ 
 $\downarrow$ 
 $L_{\text{Poly-DPO}} = -\log\left(p^{w>l}\right) + \alpha\left(1-p^{w>l}\right)$ | ```dpo_loss = -1 * logsigmoid(logits)``` 
 ```poly_loss = 1 - sigmoid(logits)``` 
 ```loss = dpo_loss + alpha * poly_loss``` | $\alpha > 0$: Noisy Dataset (Hard to learn) 
 $\alpha < 0$: Over-simple Dataset (Easy to learn) 
 $\alpha \approx 0$: High-Quality & Balanced Dataset |

Figure 3: Summary of our Poly-DPO. By adjusting only one hyperparameter and introducing only two new lines of code, our Poly-DPO can handle preference datasets with three different data distributions.

However, it is unrealistic to perturb and adjust parameters for all polynomials. A simple form is to modify only the first term that contributes the most to the gradient Leng et al. (2022), thus obtaining Poly-DPO loss:

$$L_{\text{Poly-DPO}} = -\log\left(p^{w>l}\right) + \alpha\left(1-p^{w>l}\right). \tag{9}$$

Hence, Poly-DPO rescales the DPO gradient $(1-p^{w>l})$ by $(1+\alpha p^{w>l})$, where $p^{w>l} = \sigma(\text{logit})$:

$$\frac{\partial L_{\text{Poly-DPO}}}{\partial \text{logit}} = (p^{w>l}-1) - \alpha p^{w>l}(1-p^{w>l}) = -(1-p^{w>l})\underbrace{(1+\alpha p^{w>l})}_{\text{Poly factor}}. \tag{10}$$

- $\alpha > 0$ (Confidence Enhancing). When datasets contain conflicting preference patterns, models struggle to extract consistent signals. Setting $\alpha > 0$ upweights uncertain samples (probability near 0.5) and downweights extreme cases (near 0 or 1). This prevents the model from being confused by conflicting patterns, and instead focuses learning on borderline cases where consistent improvement is possible.

- $\alpha < 0$ (Confidence Reducing). When datasets contain trivially distinguishable preferences (e.g., our synthetic dataset with shuffled losers in Section 5.2), models quickly achieve high confidence but only learn surface-level distinctions. Setting $\alpha < 0$ reduces gradient contributions from high-confidence samples, preventing over-fitting and forcing continued exploration of winner-loser differences.

- $\alpha = 0$ (Standard DPO). When datasets contain balanced, high-quality preference signals without significant conflicts or trivial patterns, the optimal configuration of Poly-DPO converges to standard DPO and is highly robust to the choice of $\alpha$.

***Remark.*** As shown in Figure 3, our Poly-DPO augments Diffusion-DPO with a *single* additive term that makes training explicitly confidence-aware. By tuning $\alpha$, it dynamically reweights samples across models and preference datasets, pushing the learning process toward informative samples while tempering over- and under-confidence, making the diffusion model better capture diverse preference patterns and achieve higher generation quality. In Section 5.2 and Figure 4, we verify the effectiveness of $\alpha$ for these three different preference distribution datasets.

## 4 LARGE-SCALE VISUAL PREFERENCE DATASET CONSTRUCTION

**Motivation and Design Principles.** Current open-source preference datasets suffer from three critical limitations that fundamentally impede scaling: (i) low resolution (512-768px) and limited prompt diversity restrict learning of fine-grained details; (ii) reliance on early-generation models produces unreliable preference signals; and (iii) random collection creates imbalanced distributions where simple patterns dominate while critical aspects remain underrepresented. To address these challenges, we construct a large-scale dataset using state-of-the-art models (FLUX, Qwen-Image for images; WanVideo, Seedance for videos) with systematic categorical organization to ensure balanced, reliable preference signals. Specifically, we construct 1M high-resolution (1024px) image preference pairs across five categories and 300K video pairs across three categories, as illustrated in Figure 2. Details on specific construction pipelines, filtering procedures, and labeling strategies are provided in the Appendix.

**ViPO-Image-1M.** We organize image preferences into five dimensions, each with 200K pairs: (1) **Aesthetics**: visual appeal and artistic merit; (2) **Text-Image Alignment**: semantic correspondence with prompts; (3) **Text Rendering**: accuracy of rendered text elements; (4) **Portrait Quality**: anatomical correctness and realism; (5) **Composition**: spatial arrangement and visual organization. For data construction, we leverage publicly available prompts from HuggingFace, employ state-of-the-art generators to create high-quality pairs, and use multiple VLMs for filtering and labeling.

**ViPO-Video-300K.** Video preferences span three dimensions, each with 100K pairs: (1) **Motion Quality**: temporal dynamics and smoothness; (2) **Video-Text Alignment**: semantic correspondence throughout temporal sequences; (3) **Visual Quality**: frame clarity and temporal consistency. We employ diverse generation strategies, including I2V based on our image dataset and T2V/T2I2V with different models to create varied preference patterns.

**Open-Source Datasets**   Due to licensing constraints of proprietary models, the original dataset version containing Seedream-3.0 and Seedance-1.0 outputs may not be publicly released. To better serve the open-source community, we provide an alternative version of ViPO that substitutes these proprietary outputs with publicly available models (FLUX.2-dev for images, Wan2.2-A14B-I2V for videos), enabling full reproducibility and broader downstream applications such as diffusion distillation for open-sourced models. Our validation confirms that the released dataset maintains comparable quality, with both SFT and Poly-DPO consistently yielding significant improvements across all evaluated models and benchmarks. Detailed dataset construction updates and comprehensive experimental results are provided in Appendix E.

## 5    EXPERIMENTS

### 5.1    EXPERIMENT SETUP

**Generation Models and Training Datasets.**   We conduct experiments on image generation using SD1.5, SDXL, SD3, and FLUX models, and video generation using Wan2.1-T2V-1.3B. For SD1.5, we train on PickaPic-v2 for fair comparison with previous methods and test on multiple datasets to evaluate resilience to preference noise. We train all image models on our ViPO-Image-1M dataset (excluding text rendering subset for SD1.5 due to its limited capabilities) and train Wan2.1-T2V-1.3B on ViPO-Video-300K.

**Evaluation Protocol.**   For SD1.5, we follow established protocols using CLIP-based reward models (ImageReward Xu et al. (2023), HPSv2.1 Wu et al. (2023a), Aesthetic Predictor Schuhmann et al. (2022)) and test datasets (HPSv2 Wu et al. (2023a), Pick-a-Pic Kirstain et al. (2023), Parti Yu et al. (2022)). For high-resolution models (SDXL, SD3, FLUX), we adopt multi-dimensional evaluation: (1) Aesthetics: DeQA You et al. (2025); (2) Alignment: DPG-Bench Hu et al. (2024); (3) Text Rendering: CVTG-2K Du et al. (2025); (4) Human Quality: GPT-4o evaluation; (5) Composition: GenEval Ghosh et al. (2023). Video generation is evaluated on VBench2.0 Huang et al. (2024).

### 5.2    ABLATION STUDIES FOR POLY-DPO

To comprehensively demonstrate that our proposed Poly-DPO can adapt to different preference datasets by adjusting the single hyperparameter $\alpha$, we conduct a series of ablation experiments based on the SD1.5 model. For these experiments, we randomly sampling 300 prompts from each of four sources: the test set of the Parti dataset, the test set of Pick-a-pic V2, the test set of HPD v2, and the "Validation Unique" set of Pick-a-pic V1. This resulted in a total of 1,200 prompts, for which a single image was generated for each. These prompts are then used to simulate three scenarios with distinct characteristics as discussed in Section 3.2: (1) noisy dataset with conflicting preference patterns, (2) over-simple dataset dominated by simple preference patterns, and (3) high-quality datasets with balanced preference distributions.

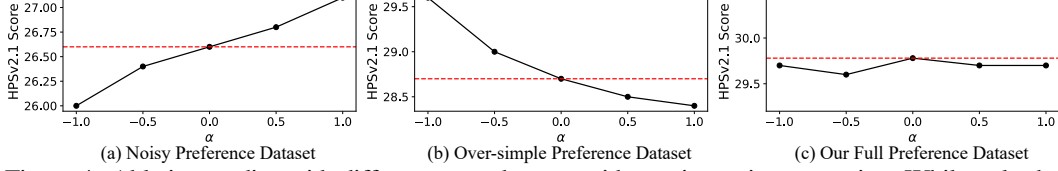

(a) Noisy Preference Dataset        (b) Over-simple Preference Dataset        (c) Our Full Preference Dataset

Figure 4: Ablation studies with different $\alpha$ on datasets with varying noise properties. While only the HPSv2.1 score is visualized for clarity, a similar trend is observed across all other evaluation metrics

**Noisy Preference Dataset.**   As the largest publicly available preference dataset, Pick-a-Pic V2 exhibits significant multi-dimensional conflicts in preference signals. Specifically, when we evaluate image pairs using five different reward models (PickScore, ImageReward, HPSv2, Aesthetic Score, and CLIP Score), only 20.79% of pairs show consistent preference rankings across all five dimensions, where one image consistently scores higher than the other. This dimensional conflict prevent models from learning meaningful preference patterns, as illustrated in Figure 4 (a). Consequently, this dataset benefits from Poly-DPO with $\alpha > 0$, which enables the model to better navigate these conflicting signals by adaptively weighting samples based on prediction confidence. In our experiments, we found $\alpha = 8$ has the best experimental results.

**Over-simple Preference Dataset.**   To validate that Poly-DPO with $\alpha < 0$ mitigates overconfidence, we construct a synthetic dataset where simple patterns dominate. We first perform SFT on SD1.5 using winner images from ViPO-Image-1M, then create preference pairs by randomly shuffling losers within batches while maintaining original winners. This setup causes a critical failure under standard DPO: the model quickly becomes overconfident and overfits to reproducing winner images rather than learning winner-loser distinctions. The high confidence from trivial preference patterns prevents the model from learning subtle preferences essential for alignment. We show that Poly-DPO with $\alpha < 0$ can penalize overconfident predictions, and forcing the model to learn more meaningful preference patterns in this scenario in Figure 4 (b).

**High-quality and Balanced Preference Dataset.** While Poly-DPO with $\alpha > 0$ and $\alpha < 0$ performs well on noisy and imbalanced datasets respectively, we observe an interesting phenomenon when training SFT-initialized SD1.5 on our complete ViPO-Image-1M dataset: the optimal $\alpha$ value converges to approximately zero, where Poly-DPO converges to standard DPO and exhibits robust performance across different hyperparameter settings, as demonstrated in Figure 4 (c). This convergence validates our dataset quality—when preferences are reliable and balanced, adaptive optimization becomes unnecessary, confirming that data quality remains the primary factor for successful and scalable preference optimization.

## 5.3 RESULTS ON PICK-A-PIC V2 TRAINING DATASET

To validate Poly-DPO's effectiveness, we conduct experiments using SD1.5 and SDXL trained on Pick-a-Pic V2, which contains substantial conflicting preference patterns as analyzed in Section 5.2, making it an ideal testbed for demonstrating robustness to noisy real-world data. Table 2 presents evaluation results across four test datasets. Poly-DPO consistently outperforms both Diffusion-DPO and Diffusion-KTO across all metrics. On Pick-a-Pic V2 test set, Poly-DPO achieves 4.4% improvement in PickScore and 13.1% in HPSv2.1, significantly surpassing Diffusion-DPO's gains of 1.8% and 4.4% respectively. The most substantial improvements appear in ImageReward scores (+0.594 vs. +0.212). This pattern holds across other test sets: on HPD V2, Poly-DPO achieves 15.9% HPSv2.1 improvement versus Diffusion-DPO's 5.3%; on Parti, the ImageReward gain reaches +0.542 versus +0.158. These consistent improvements confirm Poly-DPO's ability to extract meaningful preference signals despite conflicting patterns. Table 3 evaluates compositional understanding using GenEval benchmark. Poly-DPO achieves the highest overall scores among off-policy methods for both SD1.5 (49.87) and SDXL (60.34), even surpassing on-policy SPO while avoiding iterative sampling costs. Notably, Poly-DPO excels at challenging tasks: for SD1.5, it achieves 51.25 on counting (vs. Diffusion-DPO's 38.75) and 14.00 on attribute binding (vs. 3.75); for SDXL, attribute binding reaches 31.00 compared to Diffusion-DPO's 18.50. These substantial gains demonstrate that confidence-based reweighting enables learning nuanced preference patterns beyond simple visual attributes.

Table 2: SD1.5 comparison results when trained on the Pick-a-Pic V2 dataset and evaluated on multiple datasets. For each prompt, we generate 4 images and report the average reward scores. Baseline results are evaluated with official released checkpoints, and all evaluations are conducted under the same setting.

| Eval Dataset | Method | Paradigm | PickScore ↑ | HPSv2.1 ↑ | Aesthetic ↑ | ImageReward ↑ |
|---|---|---|---|---|---|---|
| Pick-a-Pic V2 (Test) | SD1.5 | - | 20.57 | 25.02 | 5.42 | 0.085 |
| | Diffusion-DPO | Off-Policy | $20.95_{+1.8\%}$ | $26.12_{+4.4\%}$ | $5.55_{+2.4\%}$ | $0.297_{+0.212}$ |
| | Diffusion-KTO | Off-Policy | $21.06_{+2.4\%}$ | $28.06_{+12.2\%}$ | $5.66_{+4.4\%}$ | $0.628_{+0.543}$ |
| | **Poly-DPO (Ours)** | **Off-Policy** | $\mathbf{21.48}_{+4.4\%}$ | $\mathbf{28.30}_{+13.1\%}$ | $\mathbf{5.67}_{+4.6\%}$ | $\mathbf{0.679}_{+0.594}$ |
| HPD V2 (Test) | SD1.5 | - | 20.86 | 0.246 | 5.58 | 0.139 |
| | Diffusion-DPO | Off-Policy | $21.31_{+2.2\%}$ | $0.259_{+5.3\%}$ | $5.71_{+2.3\%}$ | $0.338_{+0.199}$ |
| | Diffusion-KTO | Off-Policy | $21.45_{+2.8\%}$ | $0.284_{+15.4\%}$ | $5.80_{+3.9\%}$ | $0.690_{+0.551}$ |
| | **Poly-DPO (Ours)** | **Off-Policy** | $\mathbf{21.87}_{+4.8\%}$ | $\mathbf{0.285}_{+15.9\%}$ | $\mathbf{5.83}_{+4.5\%}$ | $\mathbf{0.716}_{+0.577}$ |
| Parti (Test) | SD1.5 | - | 21.28 | 0.253 | 5.36 | 0.194 |
| | Diffusion-DPO | Off-Policy | $21.52_{+1.1\%}$ | $0.261_{+3.2\%}$ | $5.44_{+1.5\%}$ | $0.352_{+0.158}$ |
| | Diffusion-KTO | Off-Policy | $21.59_{+1.5\%}$ | $0.279_{+10.3\%}$ | $5.55_{+3.5\%}$ | $0.615_{+0.421}$ |
| | **Poly-DPO (Ours)** | **Off-Policy** | $\mathbf{21.89}_{+2.9\%}$ | $\mathbf{0.280}_{+10.7\%}$ | $\mathbf{5.56}_{+3.7\%}$ | $\mathbf{0.736}_{+0.542}$ |
| Pick-a-Pic V1 (Validation Unique) | SD1.5 | - | 20.56 | 24.05 | 5.47 | 0.008 |
| | DDPO | On-Policy | $21.06_{+2.4\%}$ | $24.91_{+3.6\%}$ | $5.59_{+2.2\%}$ | $0.082_{+0.074}$ |
| | D3PO | On-Policy | $20.76_{+1.0\%}$ | $23.97_{-0.3\%}$ | $5.53_{+1.1\%}$ | $-0.124_{-0.132}$ |
| | SPO | On-Policy | $21.22_{+3.2\%}$ | $25.83_{+7.4\%}$ | $\mathbf{5.93}_{+8.4\%}$ | $0.168_{+0.160}$ |
| | Diffusion-DPO | Off-Policy | $20.99_{+2.1\%}$ | $25.54_{+6.2\%}$ | $5.60_{+2.4\%}$ | $0.302_{+0.294}$ |
| | Diffusion-KTO | Off-Policy | $21.12_{+2.7\%}$ | $28.19_{+17.2\%}$ | $\mathbf{5.68}_{+3.8\%}$ | $0.642_{+0.634}$ |
| | **Poly-DPO (Ours)** | **Off-Policy** | $\mathbf{21.48}_{+4.5\%}$ | $\mathbf{28.32}_{+17.8\%}$ | $\mathbf{5.68}_{+3.8\%}$ | $\mathbf{0.671}_{+0.663}$ |

## 5.4 RESULTS ON VIPO-IMAGE-1M TRAINING DATASET

**Composition.** Table 4 demonstrates the effectiveness of our ViPO-Image-1M dataset across multiple model architectures on the GenEval benchmark. All models show substantial improvements when trained with our dataset. SD1.5 improves from 0.42 to 0.52 overall (+23.8%), with particularly strong gains in two-object generation (0.38→0.66) and attribute binding (0.05→0.12). SDXL achieves 0.63 overall score, surpassing many baseline models, with attribute binding improving dramatically from 0.16 to 0.42. SD3.5-Medium, already strong at 0.69, reaches 0.83 after training, approaching the performance of HiDream-I1-Full (0.83), a model specifically designed for compositional generation. FLUX.1-dev shows consistent improvements across all metrics, reaching 0.79 overall score.

Table 3: Evaluation results on GenEval (Ghosh et al., 2023) with **Pick-a-pic V2 training dataset**. The SD1.5/SDXL/KTO/Diffusion-DPO results are evaluated with their officially released models under the same setting as LPO Zhang et al. (2025b). The SPO/LPO/MAPO baseline results are from the LPO paper.

| Model | RL Paradigm | Single Object | Two Object | Counting | Colors | Position | Attribute Binding | Overall↑ |
|---|---|---|---|---|---|---|---|---|
| SD1.5 | - | 95.62 | 37.63 | 37.81 | 74.73 | 3.50 | 4.57 | 42.34 |
| SPO | On-Policy | 95.63 | 36.62 | 34.83 | 72.34 | 3.75 | 6.50 | 41.53 |
| LPO | On-Policy | **97.81** | **55.30** | 42.19 | 80.59 | 6.75 | 10.00 | 48.77 |
| Diffusion-DPO | Off-Policy | 96.88 | 39.90 | 38.75 | 75.53 | 3.25 | 3.75 | 43.00 |
| Diffusion-KTO | Off-Policy | **97.50** | 35.35 | 36.25 | 79.79 | **7.00** | 6.00 | 43.65 |
| **Poly-DPO (Ours)** | Off-Policy | 96.25 | **46.46** | **51.25** | **87.23** | 4.00 | **14.00** | **49.87** |
| SDXL | - | 98.12 | 75.25 | 43.75 | 89.63 | 11.25 | 15.75 | 55.63 |
| SPO | On-Policy | 96.88 | 69.70 | 37.19 | 83.51 | 9.50 | 19.75 | 52.75 |
| LPO | On-Policy | **99.69** | **84.34** | 43.13 | **90.43** | 13.75 | 27.75 | 59.85 |
| Diffusion-DPO | Off-Policy | **99.38** | 82.58 | **49.06** | 85.11 | 13.05 | 18.50 | 58.02 |
| MAPO | Off-Policy | 96.56 | 66.41 | 40.00 | 84.31 | 10.75 | 18.75 | 52.80 |
| **Poly-DPO (Ours)** | Off-Policy | 98.75 | **82.83** | 46.25 | **87.23** | **16.00** | **31.00** | **60.34** |

Table 4: Evaluation results on GenEval Ghosh et al. (2023) with our **ViPO-Image-1M training dataset**.

| Model | Single Object | Two Object | Counting | Colors | Position | Attribute Binding | Overall↑ |
|---|---|---|---|---|---|---|---|
| PixArt-$\alpha$ | 0.98 | 0.50 | 0.44 | 0.80 | 0.08 | 0.07 | 0.48 |
| SD3.5 Large | 0.98 | 0.89 | 0.73 | 0.83 | 0.34 | 0.47 | 0.71 |
| HiDream-I1-Full | 1.00 | 0.98 | 0.79 | 0.91 | 0.60 | 0.72 | 0.83 |
| SD1.5 | 0.96 | 0.38 | 0.38 | 0.75 | 0.04 | 0.05 | 0.42 |
| + SFT | **0.99** | 0.49 | 0.38 | 0.78 | 0.06 | 0.09 | 0.46 |
| + SFT & Poly-DPO | 0.98 | **0.66** | **0.50** | **0.84** | **0.07** | **0.17** | **0.54** |
| SDXL | 0.98 | 0.75 | 0.44 | 0.90 | 0.11 | 0.16 | 0.56 |
| + SFT | 0.98 | 0.77 | 0.43 | 0.88 | **0.13** | 0.21 | 0.57 |
| + SFT & Poly-DPO | **1.00** | **0.88** | **0.45** | **0.93** | 0.09 | **0.42** | **0.63** |
| SD3.5-Medium | 1.00 | 0.87 | 0.68 | 0.80 | 0.20 | 0.57 | 0.69 |
| + SFT | 1.00 | 0.97 | 0.74 | 0.91 | 0.43 | 0.77 | 0.80 |
| + SFT & Poly-DPO | 1.00 | **0.97** | **0.75** | **0.91** | **0.47** | **0.86** | **0.83** |
| FLUX.1 [Dev] | **1.00** | 0.86 | 0.80 | 0.78 | 0.25 | 0.45 | 0.69 |
| + SFT | **1.00** | 0.90 | 0.74 | **0.87** | 0.38 | 0.62 | 0.75 |
| + SFT & Poly-DPO | 0.99 | **0.97** | **0.83** | 0.85 | **0.40** | **0.70** | **0.79** |

Table 5: Evaluation results on DPG-Bench Hu et al. (2024) with our **ViPO-Image-1M training dataset**.

| Model | Global | Entity | Attribute | Relation | Other | Overall↑ |
|---|---|---|---|---|---|---|
| Hunyuan-DiT | 84.59 | 80.59 | 88.01 | 74.36 | 86.41 | 78.87 |
| PixArt-$\Sigma$ | 86.89 | 82.89 | 88.94 | 86.59 | 87.68 | 80.54 |
| DALL-E 3 | 90.97 | 89.61 | 88.39 | 90.58 | 89.83 | 83.50 |
| SD3 Medium | 87.90 | 91.01 | 88.83 | 80.70 | 88.68 | 84.08 |
| HiDream-I1-Full | 76.44 | 90.22 | 89.48 | 93.74 | 91.83 | 85.89 |
| GPT-Image 1 | 88.89 | 88.94 | 89.84 | 92.63 | 90.96 | 85.15 |
| SD3.5-Medium | 91.70 | 90.59 | 89.49 | 92.21 | 85.12 | 84.24 |
| +SFT | 84.80 | 89.97 | 88.14 | 93.69 | 82.00 | 84.24 |
| +SFT & Poly-DPO | **84.80** | **92.64** | **90.10** | **94.81** | **89.20** | **87.71** |
| FLUX.1 [Dev] | 74.35 | 90.00 | 88.96 | 90.87 | 88.33 | 83.84 |
| +SFT | 85.41 | 89.21 | 85.17 | 92.72 | 80.40 | 83.59 |
| +SFT & Poly-DPO | **90.99** | **91.05** | **90.91** | **93.73** | **91.12** | **87.31** |

**Image-Text Alignment.** Tables 5 present evaluation results on text-image alignment. On DPG-Bench, both SD3.5-Medium and FLUX.1-dev achieve state-of-the-art performance after training, with overall

Table 6: SD3.5-Medium & FLUX-dev comparison results when trained on our ViPO-Image-1M dataset and evaluated across multiple benchmarks. For each prompt, we generate 4 images and report the average score. We provide more details about these experiments in the Supplementary Material.

| Method | Aesthetics DeQA ↑ | Alignment DPG-Bench ↑ | Text Rendering CVTG-2K ↑ | Human Quality GPT-4o Acc ↑ | Composition GenEval ↑ |
|---|---|---|---|---|---|
| SD3.5-Medium | 4.27 | 84.24 | 0.4378 | 73.25 | 0.69 |
| + SFT | 4.31 | 84.24 | 0.5887 | 77.50 | 0.80 |
| + SFT & Poly-DPO | **4.31** | **87.71** | **0.6995** | **85.25** | **0.83** |
| FLUX.1-dev | 4.37 | 83.84 | 0.4878 | 80.00 | 0.69 |
| + SFT | 4.32 | 83.59 | 0.2126 | 81.75 | 0.75 |
| + SFT & Poly-DPO | **4.40** | **87.31** | **0.6859** | **88.75** | **0.79** |

Table 7: Wan2.1-T2V-1.3B Experiments on VBench-2.0 when trained with our ViPO-Video-300K dataset.

| Models | Human Identity | Material | Thermotics | Dynamic Spatial Rel. | Dynamic Attribute | Motion Order Und. | Human Interaction | Camera Motion | Motion Rationality |
|---|---|---|---|---|---|---|---|---|---|
| Wan2.1 | 62.18 | 69.75 | **72.26** | 24.64 | 53.48 | 35.35 | 74.00 | 31.79 | 43.68 |
| + Poly-DPO | **67.99** | **71.57** | 68.53 | **33.82** | **57.00** | **38.62** | **78.00** | **32.49** | **47.70** |

scores of 87.71 and 87.31 respectively, surpassing commercial models like GPT-Image 1 (85.15) and approaching HiDream-I1-Full (85.89). The models excel particularly in relational understanding, with SD3.5-Medium achieving 94.81 on relation tasks.

**Aesthetics and Human Quality Evaluation.** We evaluate aesthetic quality and human generation capabilities as shown in Table 6. For aesthetic assessment using DeQA You et al. (2025), we observe modest but consistent improvements (SD3.5-Medium: 4.27→4.31, FLUX: 4.37→4.40) on DrawBench, demonstrating that our training maintains aesthetic quality while improving technical capabilities. For human quality evaluation, we use GPT-4o to assess anatomical correctness on 400 human-related prompts. The results show substantial improvements: SD3.5-Medium's accuracy increases from 73.25% to 85.25%, while FLUX improves from 80.00% to 88.75%. These gains address persistent challenges in human image generation, including correct proportions, realistic poses, and proper body structure. Our proposed ViPO-Image-1M achieves simultaneous improvements across multiple visual dimensions.

**Text Rendering.** Training on our dataset significantly improves performance on the challenging CVTG-2K text rendering benchmark Du et al. (2025). As shown in Table 6, our full pipeline boosts SD3.5-Medium's word accuracy by 59.8% (from 0.4378 to 0.6995). Notably for FLUX.1-dev, it overcomes an SFT-induced performance degradation to achieve a strong final score of 0.6859. A more detailed analysis, including results on multi-region text, is available in Tabe 15 in the Appendix.

### 5.5 RESULTS ON ViPO-VIDEO-300K TRAINING DATASET

We evaluate the effectiveness of our ViPO-Video-300K dataset using Wan2.1-T2V-1.3B model on VBench-2.0 benchmark Zheng et al. (2025), as shown in Table 7. Training with ViPO-Video-300K yields consistent improvements across nearly all evaluated dimensions. Most notably, the model shows significant gains in motion-related metrics: Dynamic Spatial Relationship improves from 24.64 to 33.82 (+37.4%), Motion Order Understanding increases from 35.35 to 38.62, and Motion Rationality rises from 43.68 to 47.70. These improvements demonstrate that our video preference dataset effectively captures temporal dynamics and motion quality distinctions. Human-centric metrics show substantial improvements, with Human Identity increasing from 62.18 to 67.99 and Human Interaction from 74.00 to 78.00, validating the quality of human motion preferences in our dataset. While Thermotics shows a slight decrease, the overall pattern of improvements across diverse evaluation criteria confirms that ViPO-Video-300K enables balanced enhancement of video generation capabilities, particularly in challenging aspects like motion understanding and temporal consistency.

### 5.6 HUMAN EVALUATION ON ViPO DATASETS

**Human Evaluation Setup.** We randomly sampled 40 images and 20 videos per category, recruiting 18 annotators to identify the superior sample in each pair based on category-specific instructions, yielding 4,378 annotations. We define *rater accuracy* as alignment with the **majority vote** across raters. All raters

exceed 70% accuracy with a mean of 87.2%, confirming high annotation reliability (see Figure 8 and Figure 9 in the Supplementary).

**Reliability of ViPO Dataset Annotations.** We benchmark the VLM-based rater against human annotators using majority-vote consensus as ground truth. As shown in Figure 5, the VLM achieves an overall agreement rate of 81.2%, surpassing the average human annotator (74.7%). The VLM excels particularly on images (84.0% vs. 74.9%), driven by strong performance on *Aesthetic* (95.0%) and *Alignment* (92.5%). For videos, VLM and human performance are comparable (71.7% vs. 72.2%), though the VLM lags on *Motion Quality* (55.0% vs. 67.2%), indicating that current VLMs still struggle with fine-grained temporal dynamics.

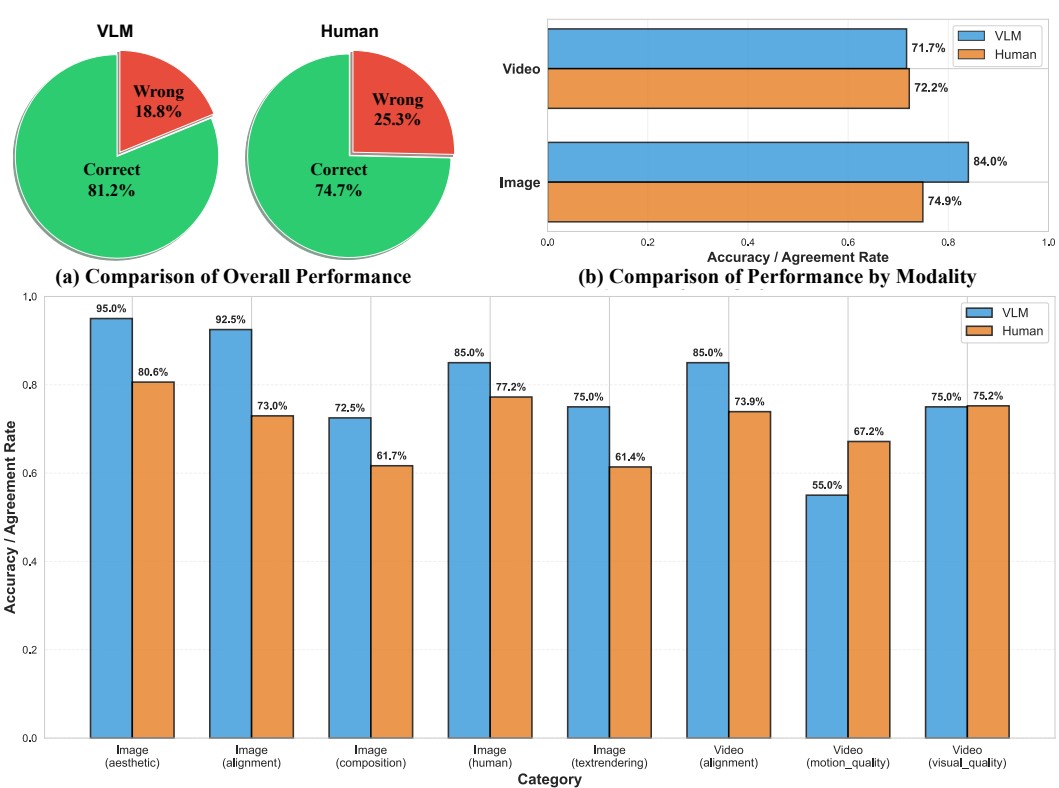

**(a) Comparison of Overall Performance**   **(b) Comparison of Performance by Modality**

**(c) Comparison of VLM and Human Performance by Category**

Figure 5: **Performance Comparison between VLM and Human Raters.** *Accuracy* (or Agreement Rate) is defined as the frequency with which a choice aligns with the consensus label (majority vote among human raters, excluding VLM predictions). **(a) Overall:** The VLM (81.2%) demonstrates higher consistency with the consensus than the average individual human annotator (74.7%). **(b) By Modality:** The VLM significantly outperforms humans on images (84.0% vs. 74.9%) but performs comparably on video tasks (71.7% vs. 72.2%). **(c) By Category:** The VLM excels in most metrics like *Aesthetic* (95.0%) but only struggles with temporal *Motion Quality* (55.0% vs. 67.2%).

## 6 CONCLUSION

In this paper, we demonstrated that conflicting preference patterns in existing datasets limit visual preference optimization scaling. We introduced Poly-DPO, which dynamically adjusts sample weighting based on confidence levels, enabling effective learning across diverse data characteristics. We also constructed ViPO, a large-scale dataset with 1M image and 300K video pairs, ensuring reliable preference signals across multiple quality dimensions. Our experiments show Poly-DPO significantly improves performance on noisy datasets like Pick-a-Pic V2 while achieving state-of-the-art results on ViPO. Remarkably, Poly-DPO converges to standard DPO on ViPO, confirming that sophisticated optimization becomes unnecessary with sufficient data quality. This reveals that scaling visual preference optimization requires addressing data quality and algorithmic robustness in tandem.

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

## A  OVERVIEW OF APPENDIX

The appendix is organized into the following sections:

- Section B: Dataset Construction Details.
- Section C: More Experiments and Analysis.
- Section D: Implementation Details.
- Section E: Results and details of the open-source datasets.
- Section F: Discussion, Limitation and Future Work.
- Section G: The Use of Large Language Models (LLMs).

## B  DATASET CONSTRUCTION DETAILS

### B.1  VIPO-IMAGE-1M DATASET

**Image-Text Alignment.**  To construct DPO preference pairs (win/loss) for image-text alignment while minimizing impact on other attributes, we utilize a single image generation model conditioned on distinct prompts to generate the corresponding image pairs.

Our data construction pipeline begins with sampling images and prompts from the open-source LAION-Aesthetics dataset. We then use Qwen2.5-VL-32B to generate a detailed caption for each image and subsequently filter out any samples containing inappropriate content. Following this, we employ Seed-VL-1.5 to perform image-grounded perturbations on these clean captions. This approach requires the model to first comprehend the image content, ensuring that all modifications are semantically consistent with the visual information. For instance, person-related attributes are only altered if human subjects are present in the image.

Specifically, we modify one, two, or three of these dimensions in the original prompt with probabilities of 70%, 20%, and 10%, respectively. The primary dimensions include: (1) style, (2) rendering, (3) lighting, (4) atmosphere, (5) time, (6) color-scheme, (7) saturation, (8) perspective, (9) depth-of-field, (10) composition, (11) weather, (12) season, (13) location, (14) background, (15) detail-level, (16) texture, (17) mood, (18) quantity, (19) size, (20) pose, (21) action, (22) interaction, (23) emotion, (24) clothing, and (25) age.

In this setup, the image generated from the original, unperturbed caption serves as the "winner", while the image generated from the perturbed caption is designated as the "loser". Based on preliminary experiments where Seedream-3.0 achieved the highest alignment scores on a small internal test set, we selected it to generate all 200K image pairs for this task.

**Text Rendering.**  The text prompts used for our text rendering dataset are constructed from three primary sources. The first component consists of 208K prompts from the CoverBook subset of the TextAtlas5M dataset. The second is a collection of 100K prompts from the 'stzhao/movie_posters_100k_controlnet' dataset on HuggingFace. The third source comprises prompts selected from the LAION-Aesthetics dataset that correspond to images containing visible text; we ensure these samples do not overlap with those used for the aforementioned image-text alignment task when sampling from LAION-Aesthetics.

After aggregating these text-centric prompts, we filter them by character count to exclude excessively long or short text strings and perform an additional step to remove inappropriate content. This process yields a final set of 200K prompts dedicated to text rendering. To construct the corresponding image pairs for these prompts, we exclusively employ Qwen-Image, HiDream-I1, Seedream-3.0, and FLUX.1-dev, as other generative models exhibit inferior text rendering capabilities.

To annotate the preference pairs for the text rendering task, we implement a two-stage evaluation process involving PaddleOCR-3.0 and Seed-VL-1.5. First, we use PaddleOCR-3.0 for an initial assessment. If one image in a pair accurately renders the text specified in the prompt while the other contains character-level errors, the former is automatically labeled as the "winner". However, if both images succeed or both fail in rendering the correct text, we proceed to the second stage. In this stage, we employ Seed-VL-1.5 to perform the comparison. The model determines the winner based on a holistic evaluation of several criteria, including the clarity of the rendered text, the precision of character formation, and the degree to which the text's position and shape align with the prompt's description.

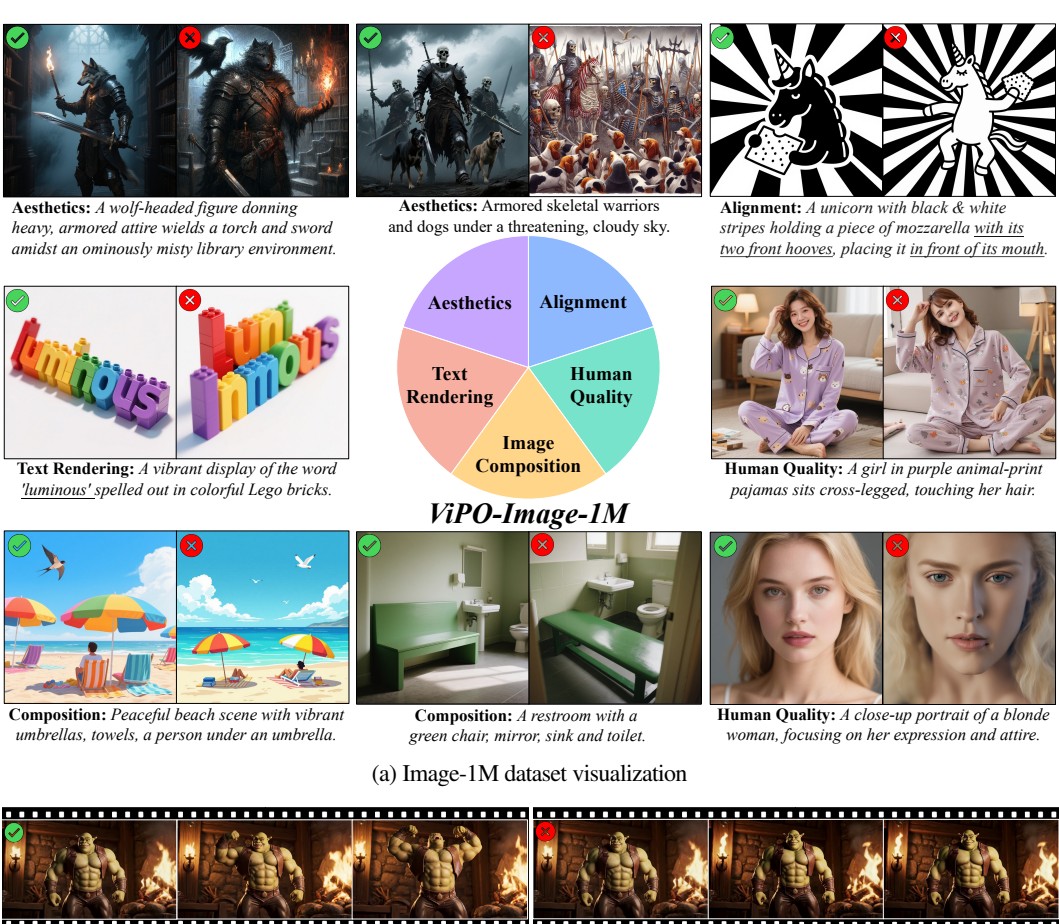

(a) Image-1M dataset visualization

(b) Video-300K dataset visualization

Figure 6: ViPO-Image-1M and ViPO-Video-300K dataset visualization.

**Human Quality.** To construct human-centric DPO dataset, we first gathered 100K images from existing open-source datasets. We began by filtering the ProGamerGov/synthetic-dataset-1m-dalle3-high-quality-captions dataset on HuggingFace with Seed-VL-1.5, selecting 44,501 images with exhibited human anatomical flaws. We augmented this dataset with 2,009 images from the gaunernst/flux-dev-portrait dataset on HuggingFace and 56,444 images from the HumanRefiner dataset on HuggingFace. This aggregated pool was then filtered for inappropriate content (e.g., violence or nudity) using Seed-VL-1.5, and finally we randomly sampled 80K images from the filtered pool.

To further diversify our dataset, we generated another 120K images. We used Seed-1.6-Lite to select 120K new human-centric prompts from the ProGamerGov/synthetic-dataset-1m-dalle3-high-quality-captions dataset, ensuring they were distinct from those used in the first step. We then prompted a suite of ten different open-source models to generate around 10K 1024x1024 images for each model (including CogView4, FLUX.1-dev, HiDream-I1-Full, Hunyuan-DiT, Kolors, PixArt-$\Sigma$, Playground-v2.5-1024px-Aesthetic, SANA1.5-4.8B-1024px, SD3.5-Medium, SDXL). In addition, we also deploy Qwen-Image to generate 20K 1024x1024 images. This resulted in a collection of 200K human-centric images sourced from a wide variety of generative models.

To create the paired preference data, we generated a counterpart for each of the 200K images using the Seedream-3.0 model with the identical prompt. Finally, Seed-VL-1.5 was employed as an automated judge to assign the final preference labels (i.e., identifying the "winner" and "loser" image in each pair) based on which image rendered the human subject more accurately. This comprehensive pipeline yielded our final dataset of 200K unique, high-resolution DPO image pairs.

**Image Composition.** We constructed our dataset by sourcing 200K unique prompts from two primary HuggingFace datasets: jackyhate/text-to-image-2M and peteromallet/high-quality-midjouney-srefs. For prompts from jackyhate/text-to-image-2M, we generated one image using Seedream-3.0 and a second, paired image using the same prompt with a randomly selected model from either Qwen-Image or HiDream-Dev. For prompts from peteromallet/high-quality-midjouney-srefs, we utilized the original MidJourney-V7 image and generated its counterpart with Seedream-3.0. Acknowledging the subjective and complex nature of evaluating image composition, we employed a multi-VLM voting system for robust preference labeling. Specifically, a panel of three diverse VLMs—Qwen2.5-VL-32B-Instruct, Seed-VL-1.5, and Q-Insight—was used to judge which image in each pair exhibited superior composition. The final preference was then determined by a majority vote from these three judges.

**Aesthetics.** To construct our aesthetics DPO dataset, we first sampled 200K prompts and corresponding images from the ProGamerGov/synthetic-dataset-1m-dalle3-high-quality-captions dataset on HuggingFace, ensuring there was no overlap with the samples previously used for the other DPO datasets. For each prompt, we generated another image using Seedream-3.0. To establish preference pairs based on aesthetics, we utilized three VLMs, i.e., Qwen2.5-VL-32B-Instruct, Seed-VL-1.5, and Q-Insight—to judge which of the two images was more aesthetically pleasing. The final preference was then determined by a majority vote.

## B.2 VIPO-VIDEO-300K

**Motion Quality.** For our Motion Quality task, we construct all video pairs using an Image-to-Video (I2V) pipeline to ensure that the spatial information between the two videos in each pair remains as consistent as possible. Our data generation process integrates samples from four distinct datasets, all sourced from HuggingFace: (1) We collect 6,763 videos and prompts from the WenhaoWang/ShareVeo3 dataset, originally generated by Veo3, extract the first frame of each, and use Seedance-1.0-Pro to synthesize 720p video pairs. (2) We take 11K prompts from LanguageBind/Open-Sora-Plan-v1.3.0, generate initial videos with HunyuanVideo-T2V-13B, extract their first frames, and then use Seedance-1.0-Lite to create the corresponding pairs. (3) We gather 32K videos from the FastVideo/Wan2.2-Syn-121x704x1280_32k dataset, generated by the WanVideo2.2 TI2V-5B model, extract the first frame and prompt for each, and use Seedance-1.0-Lite to generate the paired videos. (4) We select 50K image-text pairs from the LAION-Aesthetics dataset, augment the prompts with motion details using Seed-VL-1.5, and then generate video pairs using both Seedance-1.0-Pro and Seedance-1.0-Lite. After generating all pairs, we use Seed-VL-1.5 to score the motion quality of each video, designating the higher-scoring one as the 'winner'. We then filter the dataset by removing pairs with the largest and smallest score differences to discard trivial or ambiguous examples, resulting in a final dataset of 100K preference pairs for this task.

**Visual Quality.** To construct the Visual Quality subset of our ViPO-Video-300K dataset, we first sample 100K image pairs from ViPO-Image-1M. We specifically select samples for which all participating VLMs (Qwen2.5-VL-32B, Seed-1.5-VL, and Q-Insight) unanimously assigned the same preference label. Subsequently, for each selected pair, the images are fed into Seed-VL-1.5 to generate a single motion prompt that is semantically suitable for both. This motion prompt is then integrated with the shared image description to form the final video generation prompt. Using this prompt and the two source images, we employ Seedance-1.0-Lite to perform an image-to-video synthesis task, generating the corresponding video preference pair. The preference label for each resulting video pair is directly inherited from its source image pair.

**Video-Text Alignment.** For the Video-Text Alignment task, we construct preference data by addressing two key aspects: spatial alignment and temporal alignment. To generate data for spatial alignment, we first select 50K image-text alignment pairs from ViPO-Image-1M, which feature subtle visual differences. We then employ Seed-VL-1.5 to generate a single, common motion prompt suitable for the main subject in both images. Subsequently, Seedance-1.0-Lite executes an I2V task for each image using this shared prompt, creating video pairs where preference is determined by the inherited spatial characteristics. For temporal alignment, we select 50K images from the LAION-Aesthetics dataset. For each image, we use Seed-VL-1.5 to generate two distinct motion prompts (e.g., "a person running" vs. "a person walking"). Seedance-1.0-Lite then generates two videos from the same source image, each conditioned on one of the different motion prompts. In both scenarios, the winner-loser designation is based on the correspondence between a video and its prompt; the video that accurately reflects its conditioning prompt is the winner.

## C    MORE EXPERIMENTS AND ANALYSIS

**Detailed Text Rendering Results.** A distinctive advantage of our dataset is the significant improvement in text rendering on CVTG-2K benchmark Du et al. (2025), which is a historically challenging task for diffusion models. As shown in Table 15, SD3.5-Medium's average word accuracy improves from 0.4378 to 0.6995 (+59.8%), with the NED score reaching 0.8853. FLUX.1-dev demonstrates even more dramatic gains, improving from 0.4878 to 0.6859 in word accuracy despite SFT alone causing degradation (0.2126). These improvements are particularly notable for multi-region text rendering, where SD3.5-Medium achieves 0.6252 accuracy on 5-region text compared to the baseline's 0.3933.

Table 8: Quantitative evaluation results of English text rendering on CVTG-2K Du et al. (2025).

| Model | Word Accuracy↑ | | | | | NED↑ | CLIPScore↑ |
|---|---|---|---|---|---|---|---|
| | 2 regions | 3 regions | 4 regions | 5 regions | average | | |
| SD3.5 Large | 0.7293 | 0.6825 | 0.6574 | 0.5940 | 0.6548 | 0.8470 | 0.7797 |
| AnyText | 0.0513 | 0.1739 | 0.1948 | 0.2249 | 0.1804 | 0.4675 | 0.7432 |
| TextDiffuser-2 | 0.5322 | 0.3255 | 0.1787 | 0.0809 | 0.2326 | 0.4353 | 0.6765 |
| RAG-Diffusion | 0.4388 | 0.3316 | 0.2116 | 0.1910 | 0.2648 | 0.4498 | 0.7797 |
| 3DIS | 0.4495 | 0.3959 | 0.3880 | 0.3303 | 0.3813 | 0.6505 | 0.7767 |
| TextCrafter | 0.7628 | 0.7628 | 0.7406 | 0.6977 | 0.7370 | 0.8679 | 0.7868 |
| SD3.5-Medium | 0.5104 | 0.4788 | 0.4197 | 0.3933 | 0.4378 | 0.7325 | 0.7548 |
| +SFT | 0.7474 | 0.6485 | 0.5625 | 0.5027 | 0.5887 | 0.8228 | 0.8107 |
| +SFT & Poly-DPO | **0.8188** | **0.7422** | **0.6900** | **0.6252** | **0.6995** | **0.8853** | **0.8287** |
| FLUX.1 [dev] | 0.6532 | 0.5273 | 0.4491 | 0.4312 | 0.4878 | 0.6727 | 0.7265 |
| +SFT | 0.3530 | 0.2462 | 0.1962 | 0.1459 | 0.2126 | 0.4623 | 0.7303 |
| +SFT & Poly-DPO | **0.7733** | **0.7203** | **0.6893** | **0.6169** | **0.6859** | **0.8489** | **0.7939** |

**Supervised Fine-Tuning on ViPO-Image-1M.** The results presented in Table 9 highlight the optimal strategy for integrating Supervised Fine-Tuning (SFT) with our Poly-DPO method. All models in this ablation are evaluated on the same 1,200-prompt test set detailed in Section 5.2. We first observe that an initial SFT stage is crucial for achieving the best performance. Applying Poly-DPO directly to the SD1.5 baseline yields only modest improvements, whereas models that first undergo SFT before DPO training demonstrate substantially higher scores across all evaluation metrics.

Furthermore, our experiments reveal that the composition of the SFT dataset is critical. By comparing models trained with SFT on the full winner-loser pairs versus only the winner images, we consistently find that the latter approach is superior. This is evidenced by our top-performing model, "+ SFT (Winner Only)

& Poly-DPO," which surpasses all other configurations. This demonstrates that fine-tuning exclusively on high-preference (winner) data provides a more effective foundation for the subsequent preference alignment with Poly-DPO.

Table 9: Ablation study on the integration of Supervised Fine-Tuning (SFT) and Poly-DPO for the SD1.5 model. The results demonstrate that an initial SFT stage using only winner images is the optimal strategy to achieve the best performance. We utilize this optimal setting for all experiments in the main paper.

| Method | PickScore ↑ | HPSv2.1 ↑ | Aesthetic ↑ | ImageReward ↑ |
|---|---|---|---|---|
| SD1.5 | 20.89 | 25.04 | 5.46 | 0.1757 |
| + Poly-DPO | 21.51 | 26.40 | 5.60 | 0.6391 |
| + SFT (Winner-Loser) | 21.74 | 28.75 | 5.71 | 0.7671 |
| + SFT (Winner Only) | 21.92 | 29.00 | 5.72 | 0.8355 |
| + SFT (Winner-Loser) & Poly-DPO | 22.06 | 29.57 | 5.76 | 0.9955 |
| + SFT (Winner Only) & Poly-DPO | **22.19** | **29.69** | **5.78** | **1.0161** |

**Gradient Analysis on $\alpha$ of Our Poly-DPO.** Figure 7 visualizes how the gradient magnitude $|\frac{\partial L}{\partial z}| = |-(1-p)(1+\alpha p)|$ of Poly-DPO adapts to different data characteristics through the $\alpha$ parameter, where $p = \sigma(z)$ represents the model's confidence in preferring the chosen response. The visualization reveals three distinct optimization regimes that directly correspond to our experimental findings. When $\alpha > 0$ (blue and purple curves), the gradient is amplified in the region $p \in [0.5, 0.8]$, maintaining substantial parameter updates even for moderately confident predictions. This enhancement proves crucial for noisy datasets like Pick-a-Pic V2, where only 20.79% of samples show consistent preferences across evaluation dimensions—the sustained gradient (approximately 2-3× stronger than standard DPO at $p \approx 0.6$ when $\alpha = 8$) prevents premature convergence on spurious patterns and encourages continued exploration to identify genuine preference signals amidst dimensional conflicts. Conversely, when $\alpha < 0$ (red and orange curves), the gradient decays more rapidly as confidence increases, actively penalizing overconfident predictions. This mechanism addresses the overconfidence problem in our synthetic dataset experiment, where negative $\alpha$ values enforce faster gradient decay beyond $p > 0.6$, maintaining the model in a "humble" learning state that prevents memorization of superficial patterns. Remarkably, when training on our high-quality ViPO-Image-1M dataset, the optimal $\alpha$ converges to approximately zero (green curve), where Poly-DPO reduces to standard DPO with linear gradient decay $|-(1-p)|$. This convergence serves as an empirical validation of dataset quality—when preference labels are reliable and balanced, additional gradient modulation becomes unnecessary, confirming that data quality remains fundamental for successful preference optimization. The visualization also provides practical insights: the optimal $\alpha$ value serves as a diagnostic tool for dataset quality (large positive values suggest noisy labels, negative values indicate oversimplified patterns, while $\alpha \approx 0$ validates well-balanced data), and explains why different datasets achieve different convergence points. This adaptive gradient mechanism enables Poly-DPO to achieve robust performance across diverse dataset characteristics without requiring dataset-specific algorithmic modifications.

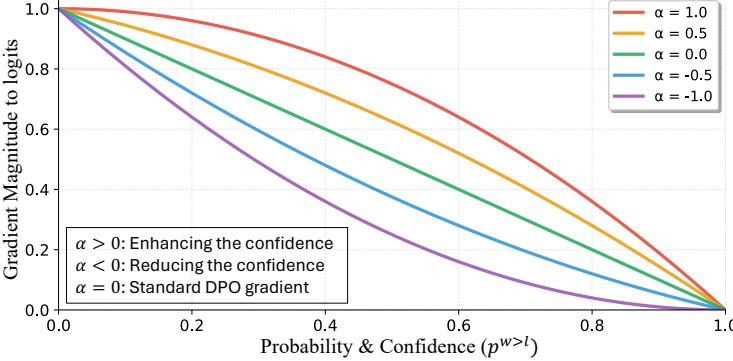

Figure 7: Gradient magnitude of Poly-DPO loss with respect to logits under different $\alpha$ values. The gradient $|-(1-p)(1+\alpha p)|$ adaptively controls learning dynamics based on confidence $p$. $\alpha > 0$ enhances gradients for medium-confidence predictions to combat noisy labels, $\alpha < 0$ suppresses overconfident predictions to prevent overfitting, while $\alpha = 0$ (standard DPO) proves optimal for high-quality balanced datasets.

**Human Evaluation Details**    To ensure the rigorous quality standards of the ViPO dataset, we conducted a large-scale evaluation by recruiting 18 annotators. This scale significantly exceeds that of related visual generation works, such as ControlNet (12), thereby offering higher statistical confidence and mitigating individual bias. Figure 8 details the rater reliability, defined as the consistency between an individual's choices and the majority vote consensus. The empirical results highlight exceptional agreement: every rater surpassed $70\%$ accuracy, with 14 out of 18 exceeding $80\%$ (Mean: $87.2\%$, Median: $87.6\%$). This distribution confirms that our collected preference labels are stable and trustworthy. Such high inter-rater agreement further evidences that the ViPO tasks are well-posed and the instructions are unambiguous, effectively minimizing the noise often inherent in subjective visual assessments. Consequently, the derived consensus labels provide a robust ground truth for benchmarking. Finally, Figure 9 illustrates the annotation interface; rater IDs are utilized strictly for tracking and resuming management to guarantee a fully anonymous evaluation process.

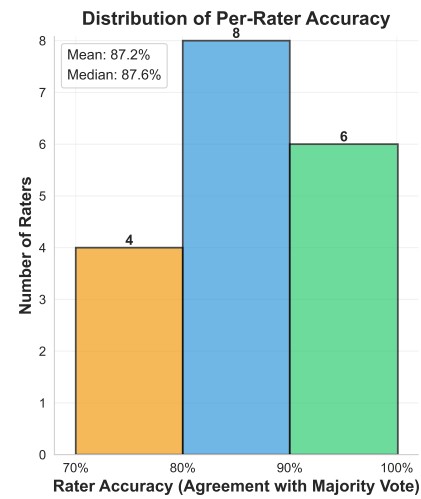

Figure 8: Distribution of human rater accuracy on our ViPO datasets.

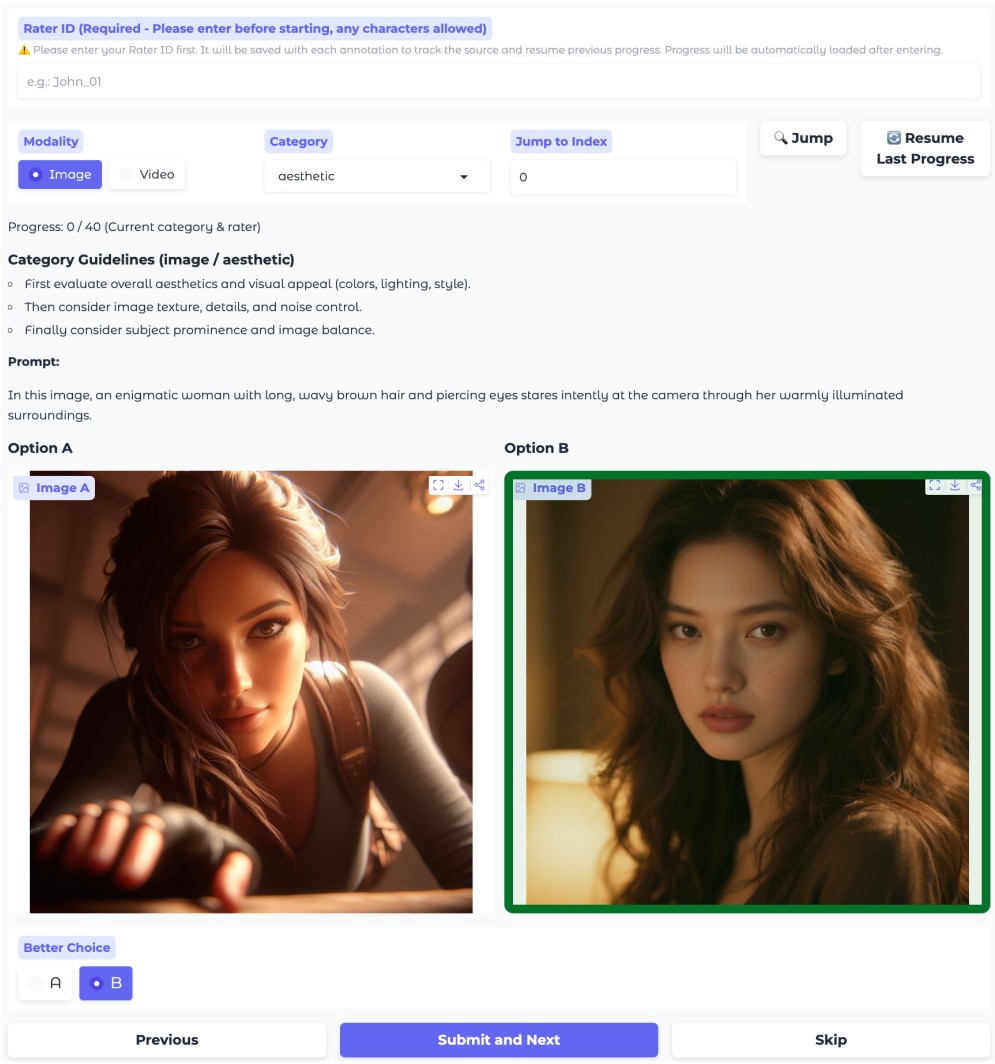

Figure 9: The UI inferface used for our human evaluation.

**SD1.5 & SDXL Experiments on DPG-Bench.** Table 10 presents the comparative results on the DPG-Bench benchmark. As shown, our proposed Poly-DPO consistently outperforms existing baselines across both SD1.5 and SDXL backbones, achieving the highest Overall scores of 67.02 and 75.67, respectively. This demonstrates the superior capability of our Poly-DPO in aligning diffusion models with human preferences. regarding the baseline selection, it is worth noting that we report results for Diffusion-KTO exclusively on SD1.5 and MAPO on SDXL, as their respective official repositories have only released model weights for these specific architectures.

Table 10: Evaluation results on DPG-Bench Hu et al. (2024) with the Pick-a-pic V2 training dataset

| Model | Paradigm | Global | Entity | Attribute | Relation | Other | Overall↑ |
|---|---|---|---|---|---|---|---|
| SD1.5 Rombach et al. (2022) | Off-Policy | **74.63** | 74.23 | 75.39 | 73.49 | 67.81 | 63.18 |
| Diffusion-DPO Rafailov et al. (2023) | Off-Policy | 71.50 | 72.53 | 75.25 | 73.55 | 72.84 | 63.29 |
| Diffusion-KTO Li et al. (2024b) | Off-Policy | 72.45 | 76.51 | **78.09** | **78.08** | 73.20 | 66.69 |
| Poly-DPO | Off-Policy | 73.36 | **78.15** | 76.50 | 75.81 | **73.42** | **67.02** |
| SDXL Podell et al. (2023) | Off-Policy | 83.27 | 82.43 | 80.91 | **86.76** | 80.41 | 74.65 |
| Diffusion-DPO Rafailov et al. (2023) | Off-Policy | 83.67 | 83.50 | **81.89** | 81.56 | **81.58** | 75.12 |
| MAPO Hong et al. (2024) | Off-Policy | 78.22 | 81.31 | 80.65 | 85.35 | 79.85 | 73.80 |
| Poly-DPO | Off-Policy | **84.03** | **83.86** | 81.87 | 83.07 | 81.02 | **75.67** |

**Inference on Different ViPO Sub-datasets.** Table 11 comprehensively evaluates the performance of the SD3.5-Medium model under various fine-tuning strategies, leveraging distinct sub-datasets from ViPO. Initially, the base SD3.5-Medium model serves as our benchmark, demonstrating solid performance across all metrics. The subsequent rows clearly illustrate the significant benefits of Supervised Fine-Tuning (SFT) using individual ViPO sub-datasets. For instance, SFT on the "Aesthetics" dataset noticeably improves DeQA and DPG-Bench scores, while SFT on "Text Rendering" leads to a substantial jump in CVTG-2K. This initial phase highlights the high quality and specificity of our ViPO sub-datasets, as targeted training on specific aspects like aesthetics or text rendering yields immediate and measurable improvements in their corresponding evaluation metrics.

A crucial observation is the inherent overlap among these diverse datasets. For example, datasets primarily focused on "Aesthetics" or "Alignment" inevitably contain elements pertaining to "Human Quality" and "Text Rendering." Consequently, fine-tuning on a seemingly specific dataset can still positively influence other, indirectly related metrics. This is evident in several SFT rows, where improvements are not strictly confined to the explicitly targeted metric. When SFT is applied to "All Datasets," we observe a more generalized enhancement, albeit with some trade-offs, indicating the complexity of balancing multiple objectives through SFT alone.

The most compelling results emerge from the combination of SFT (on "All Datasets") followed by DPO using individual ViPO sub-datasets. This two-stage approach consistently achieves superior performance across all evaluation metrics, significantly surpassing both the base model and models trained with SFT alone. Notably, the "All Datasets" DPO fine-tuning achieves the highest scores across most metrics, including a remarkable 85.25 for GPT-4o Accuracy and 0.6995 for CVTG-2K, representing a substantial leap from the SFT-only and base models. This profound improvement underscores two key points: first, the high quality and preference-rich nature of our ViPO datasets are exceptionally well-suited for preference learning; and second, DPO effectively harnesses this high-quality preference data to further refine the model's capabilities, leading to more robust and human-aligned outputs across various dimensions like aesthetics, alignment, text rendering, and overall human quality. The consistent gains across different DPO fine-tuning setups further validate the effectiveness of our comprehensive training methodology and the superior learning signals provided by the ViPO dataset.

**Inference on Different SFT Training Steps.** Table 12 presents an ablation study on the number of training steps during the SFT phase, ranging from 1,000 to 4,000 steps. As observed, extending the training duration yields a continuous and significant improvement in complex capabilities such as Text Rendering (CVTG-2K) and Human Quality (GPT-4o Acc), with the latter increasing from 73.25 to a peak of 77.50. While some metrics like Alignment (DPG-Bench) saturate or slightly fluctuate after early stages, the steady gains in text rendering (reaching 0.5887) and overall human preference indicate that the model requires more training steps to fully absorb the fine-grained details present in our high-quality dataset. Consequently, we select the 4,000-step checkpoint for subsequent stages, as it offers the most robust foundation for generating high-fidelity, human-preferred images.

Table 11: SD3.5-Medium performance on various metrics after fine-tuning with different ViPO sub-datasets using SFT and DPO, demonstrating the impact of specific and comprehensive data training.

| Method | Dataset | Aesthetics DeQA ↑ | Alignment DPG-Bench ↑ | Text Rendering CVTG-2K ↑ | Human Quality GPT-4o Acc ↑ | Composition GenEval ↑ |
|---|---|---|---|---|---|---|
| SD3.5-Medium | - | 4.27 | 84.24 | 0.4378 | 73.25 | 0.69 |
| + SFT | Aesthetics | 4.32 | 87.02 | 0.5051 | 76.91 | 0.78 |
| + SFT | Alignment | 4.30 | 86.63 | 0.4904 | 76.89 | 0.77 |
| + SFT | Composition | 4.30 | 86.57 | 0.4815 | 76.52 | 0.78 |
| + SFT | Human Quality | 4.29 | 87.05 | 0.5174 | 77.42 | 0.79 |
| + SFT | Text Rendering | 4.25 | 85.85 | 0.5319 | 74.45 | 0.76 |
| + SFT | All Datasets | 4.31 | 84.24 | 0.5887 | 77.50 | 0.80 |
| + SFT (All) + DPO | Aesthetics | 4.31 | 86.91 | 0.5668 | 82.32 | 0.79 |
| + SFT (All) + DPO | Alignment | 4.31 | 88.55 | 0.6680 | 82.14 | 0.79 |
| + SFT (All) + DPO | Composition | 4.31 | 86.41 | 0.6190 | 81.78 | 0.80 |
| + SFT (All) + DPO | Human Quality | 4.30 | 86.70 | 0.5729 | 83.02 | 0.81 |
| + SFT (All) + DPO | Text Rendering | 4.28 | 86.13 | 0.6344 | 80.18 | 0.79 |
| + SFT (All) + DPO | All Datasets | 4.31 | 87.71 | 0.6995 | 85.25 | 0.83 |

Table 12: Ablation study on the effect of training steps during the Supervised Fine-Tuning (SFT) stage.

| Method | Aesthetics DeQA ↑ | Alignment DPG-Bench ↑ | Text Rendering CVTG-2K ↑ | Human Quality GPT-4o Acc ↑ | Composition GenEval ↑ |
|---|---|---|---|---|---|
| SD3.5-Medium | 4.27 | 84.24 | 0.4378 | 73.25 | 0.69 |
| + SFT 1000 Steps | 4.28 | 86.84 | 0.5134 | 73.98 | 0.79 |
| + SFT 2000 Steps | 4.31 | 86.72 | 0.5334 | 75.16 | 0.81 |
| + SFT 3000 Steps | 4.30 | 86.27 | 0.5614 | 76.34 | 0.81 |
| + SFT 4000 Steps | 4.31 | 84.24 | 0.5887 | 77.50 | 0.80 |

**Training Stability for Diffusion-DPO and Poly-DPO.**    To address concerns regarding potential model collapse, we visualize the training dynamics of both the baseline Diffusion-DPO and our proposed Poly-DPO. As illustrated in Figure 10, we track four key evaluation metrics—PickScore, ImageReward, Aesthetic Score, and HPSv2—throughout the training process on the Pick-a-Pic V2 dataset. Contrary to the instability often associated with on-policy RL methods, both off-policy approaches demonstrate remarkable stability. The reward scores exhibit a consistent, steady increase followed by a smooth plateau, indicating a stable convergence process with no signs of sudden performance degradation or collapse. Notably, Poly-DPO maintains the robust stability inherent to the DPO framework while achieving a higher performance ceiling than the baseline.

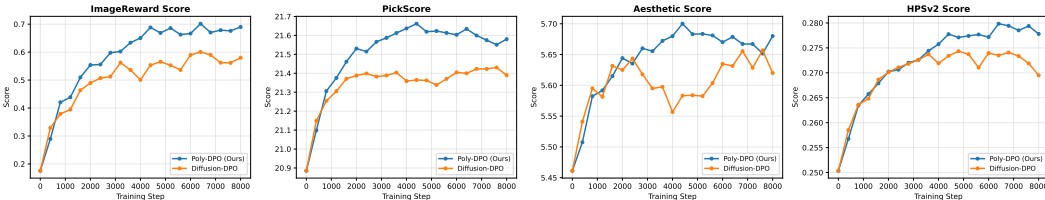

Figure 10: Training dynamics of Poly-DPO and Diffusion-DPO on the Pick-a-Pic V2 dataset. Both methods exhibit high training stability, with evaluation metrics steadily increasing to convergence without any signs of model collapse.

## D    IMPLEMENTATION DETAILS

**Training on Pick-a-pic V2 Dataset.**    We validate our proposed Poly-DPO method by training the SD1.5 model on the Pick-a-pic V2 dataset. Our training implementation and hyperparameters are based on the official open-source code of Diffusion-DPO. Specifically, we use a batch size of 512 and a base learning rate of 4e-9 (the final learning rate is $512 \times 4e-9 = 2.048e-6$), the training resolution is $512 \times 512$. We perform a grid search for the hyperparameter $\alpha$ of Poly-DPO over the set{-1, -0.5, 0, 0.5, 1, 2, 4, 6, 8, 10} and find that $\alpha = 8$ yields the best results.In addition, we observed that the original Diffusion-DPO algorithm converges in approximately 8,000 steps, whereas our Poly-DPO method achieves convergence in

4,500 steps. Throughout the training process, we do not update the reference model or use the Exponential Moving Average (EMA).

**Training on ViPO-Image-1M Dataset.** For our experiments on the ViPO-Image-1M training set, we first conduct validation on the SD1.5 model. Based on our conclusions in Section 9, we adopt a two-stage training process for all models. First, we perform SFT using only the winner images. Following this, we apply Poly-DPO training. For this second stage, it is important to note that both the policy model being trained and the reference model are initialized from the checkpoint of the SFT-tuned model. We found that there was no significant difference in the evaluation indicators when $\alpha$ was in the range of [-1, 1], for both the SD1.5 and the SDXL model, so we simply set $\alpha = 0$ for all experiments. The training resolution is $512 \times 512$ for SD1.5 and $1024 \times 1024$ for other models. No reference model update or EMA is used for all experiments. The specific implementation details for each model architecture are as follows:

- **SD1.5.** We use a batch size of 512 for both stages. The base learning rate is 4e-9 for SFT and 1e-9 for Poly-DPO, with both stages trained for 8,000 steps. We observed that after the initial SFT, a smaller value for $\beta$ in Equation 5 was better, so we set $\beta = 500$.

- **SDXL.** The batch size is 512. The base learning rates are 2e-9 for SFT and 5e-10 for Poly-DPO, with both stages trained for 4,000 steps. We use $\beta = 1000$ for this model.

- **SD3.5-Medium.** For the SFT stage, we use a batch size of 2048 and a base learning rate of 1e-8. For the Poly-DPO stage, the batch size is 512 with a base learning rate of 5e-9. The SFT stage is trained for 4,000 steps and the Poly-DPO stage for 2,000 steps, with $\beta = 500$.

- **FLUX.1-dev.** For the SFT stage, the batch size is 2048 with a base learning rate of 1e-9. For the Poly-DPO stage, the batch size is 512 with a base learning rate of 5e-9. Similar to SD3.5-Medium, SFT is trained for 4,000 steps and Poly-DPO for 2,000 steps, using $\beta = 500$.

**Training on ViPO-Video-300K Dataset.** We conduct experiments by applying Poly-DPO directly to the Wan2.1-T2V-1.3B base model, using the ViPO-Video-300K dataset for training. The model is trained for 2,000 steps with a batch size of 256 and a base learning rate of 1e-8. For this experiment, we set the DPO hyperparameter $\beta = 500$ and the Poly-DPO hyperparameter $\alpha = 0$. During training, we utilize a dynamic resolution approach and do not perform any resizing operations on the videos in the dataset. This means we consistently train on video data with its original 16:9 and 1:1 aspect ratios. For final evaluation, the VBench2.0 score is calculated by averaging the results from both the 16:9 and 1:1 generated videos.

# E OPEN-SOURCE DATASET CONSTRUCTION

## E.1 MOTIVATION

As discussed in Section 4, the original ViPO dataset leverages several proprietary generative models, including Seedream-3.0 for image generation and Seedance-1.0 for video generation. Due to licensing and intellectual property constraints associated with these proprietary models, we are unable to release the data generated by them. However, we believe that open access to high-quality visual preference data is essential for advancing research in this area. To this end, we construct and release an alternative version of the ViPO dataset that replaces all proprietary model outputs while preserving the dataset's scale, categorical structure, and preference quality.

Importantly, this replacement is limited in scope. The majority of the ViPO dataset is constructed using outputs from open-source generative models (e.g., FLUX.1-dev, HiDream-I1, Qwen-Image, WanVideo) and prompts sourced from publicly available datasets on HuggingFace. These portions of the dataset remain unchanged in the released version. Only the subsets originally generated by Seedream-3.0 (for images) and Seedance-1.0 (for videos) are substituted, with FLUX.2-dev Labs (2025) and Wan2.2-A14B-I2V Wan et al. (2025) serving as their respective replacements. Both replacement models are publicly available and demonstrate strong generation capabilities comparable to their proprietary counterparts. We name the released datasets **ViPO-Image-1M-Open** and **ViPO-Video-300K-Open**, respectively.

## E.2 ANNOTATION PIPELINE

To ensure consistent and reliable preference labels across the released dataset, we adopt a unified multi-VLM voting pipeline for all annotations, covering both image and video preference pairs. Specifically,

each candidate pair is independently evaluated by three Vision-Language Models: Qwen3-VL-30B-A3B-Thinking Bai et al. (2025), Molmo2-8B Clark et al. (2026), and Seed-1.8 Seed (2025). Each VLM is prompted with category-specific instructions (e.g., assessing aesthetic quality, text-image alignment, or motion quality) and independently selects a preferred sample from each pair. The final preference label is determined by majority vote among the three judges. This multi-VLM voting strategy mitigates individual model biases and produces robust, reproducible annotations without relying on proprietary labeling services.

## E.3 EXPERIMENTAL VALIDATION

To verify that the released dataset maintains the quality and effectiveness of the original version, we re-run the main experiments reported in the paper. Specifically, we train SD1.5, SDXL, SD3.5-Medium, and FLUX.1-dev on the released ViPO-Image-1M-Open, following the same two-stage pipeline (SFT followed by Poly-DPO) described in Section D.

Table 13: Evaluation results on GenEval Ghosh et al. (2023) with different training datasets.

| Model | Training Dataset | Single Object | Two Object | Counting | Colors | Position | Attribute Binding | Overall↑ |
|---|---|---|---|---|---|---|---|---|
| SD1.5 | - | 0.96 | 0.38 | 0.38 | 0.75 | 0.04 | 0.05 | 0.42 |
| + SFT | ViPO-Image-1M | 0.99 | 0.49 | 0.38 | 0.78 | 0.06 | 0.09 | 0.46 |
| + SFT & Poly-DPO | ViPO-Image-1M | 0.98 | 0.66 | 0.50 | 0.84 | 0.07 | 0.17 | 0.54 |
| + SFT | ViPO-Image-1M-Open | 0.99 | 0.50 | 0.49 | 0.77 | 0.02 | 0.08 | 0.47 |
| + SFT & Poly-DPO | ViPO-Image-1M-Open | 0.98 | 0.57 | 0.63 | 0.78 | 0.08 | 0.17 | 0.53 |
| SDXL | - | 0.98 | 0.75 | 0.44 | 0.90 | 0.11 | 0.16 | 0.56 |
| + SFT | ViPO-Image-1M | 0.98 | 0.77 | 0.43 | 0.88 | 0.13 | 0.21 | 0.57 |
| + SFT & Poly-DPO | ViPO-Image-1M | 1.00 | 0.88 | 0.45 | 0.93 | 0.09 | 0.42 | 0.63 |
| + SFT | ViPO-Image-1M-Open | 0.99 | 0.73 | 0.58 | 0.83 | 0.13 | 0.28 | 0.59 |
| + SFT & Poly-DPO | ViPO-Image-1M-Open | 1.00 | 0.87 | 0.49 | 0.89 | 0.15 | 0.37 | 0.63 |
| SD3.5-Medium | - | 1.00 | 0.87 | 0.68 | 0.80 | 0.20 | 0.57 | 0.69 |
| + SFT | ViPO-Image-1M | 1.00 | 0.97 | 0.74 | 0.91 | 0.43 | 0.77 | 0.80 |
| + SFT & Poly-DPO | ViPO-Image-1M | 1.00 | 0.97 | 0.75 | 0.91 | 0.47 | 0.86 | 0.83 |
| + SFT | ViPO-Image-1M-Open | 1.00 | 0.97 | 0.69 | 0.90 | 0.42 | 0.72 | 0.78 |
| + SFT & Poly-DPO | ViPO-Image-1M-Open | 1.00 | 0.99 | 0.76 | 0.90 | 0.45 | 0.71 | 0.81 |
| FLUX.1 [Dev] | - | 1.00 | 0.86 | 0.80 | 0.78 | 0.25 | 0.45 | 0.69 |
| + SFT | ViPO-Image-1M | 1.00 | 0.90 | 0.74 | 0.87 | 0.38 | 0.62 | 0.75 |
| + SFT & Poly-DPO | ViPO-Image-1M | 0.99 | 0.97 | 0.83 | 0.85 | 0.40 | 0.70 | 0.79 |
| + SFT | ViPO-Image-1M-Open | 0.99 | 0.89 | 0.80 | 0.83 | 0.29 | 0.67 | 0.74 |
| + SFT & Poly-DPO | ViPO-Image-1M-Open | 1.00 | 0.92 | 0.79 | 0.85 | 0.29 | 0.68 | 0.76 |

Table 14: Evaluation results on DPG-Bench Hu et al. (2024) with different training datasets.

| Model | Training Dataset | Global | Entity | Attribute | Relation | Other | Overall↑ |
|---|---|---|---|---|---|---|---|
| SD3.5-Medium | - | 91.70 | 90.59 | 89.49 | 92.21 | 85.12 | 84.24 |
| +SFT | ViPO-Image-1M | 84.80 | 89.97 | 88.14 | 93.69 | 82.00 | 84.24 |
| +SFT & Poly-DPO | ViPO-Image-1M | 84.80 | 92.64 | 90.10 | 94.81 | 89.20 | 87.71 |
| +SFT | ViPO-Image-1M-Open | 92.06 | 93.60 | 91.56 | 91.13 | 88.68 | 87.57 |
| +SFT & Poly-DPO | ViPO-Image-1M-Open | 94.47 | 94.08 | 93.19 | 91.86 | 93.29 | 89.85 |
| FLUX.1 [Dev] | - | 74.35 | 90.00 | 88.96 | 90.87 | 88.33 | 83.84 |
| +SFT | ViPO-Image-1M | 85.41 | 89.21 | 85.17 | 92.72 | 80.40 | 83.59 |
| +SFT & Poly-DPO | ViPO-Image-1M | 90.99 | 91.05 | 90.91 | 93.73 | 91.12 | 87.31 |
| +SFT | ViPO-Image-1M-Open | 92.21 | 91.44 | 89.47 | 91.66 | 89.51 | 85.41 |
| +SFT & Poly-DPO | ViPO-Image-1M-Open | 82.09 | 91.94 | 92.12 | 91.91 | 90.44 | 86.99 |

The results confirm that all core conclusions of the paper hold with the released dataset: (1) SFT on winner images consistently improves base model performance across all architectures and evaluation dimensions; (2) Poly-DPO further yields significant gains on top of SFT, demonstrating the effectiveness of preference optimization; and (3) the overall performance trends across benchmarks remain consistent with those reported using the original dataset. These findings validate that the released ViPO dataset is a faithful and effective substitute, suitable for reproducing our results and supporting future research in visual preference optimization.

Table 15: Quantitative evaluation results of English text rendering on CVTG-2K Du et al. (2025) with our ViPO-Image-1M and ViPO-Image-1M-Open training datasets.

| Model | Training Dataset | Word Accuracy↑ | | | | | NED↑ | CLIPScore↑ |
|---|---|---|---|---|---|---|---|---|
| | | 2 regions | 3 regions | 4 regions | 5 regions | average | | |
| SD3.5-Medium | - | 0.5104 | 0.4788 | 0.4197 | 0.3933 | 0.4378 | 0.7325 | 0.7548 |
| +SFT | ViPO-Image-1M | 0.7474 | 0.6485 | 0.5625 | 0.5027 | 0.5887 | 0.8228 | 0.8107 |
| +SFT & Poly-DPO | ViPO-Image-1M | 0.8188 | 0.7422 | 0.6900 | 0.6252 | 0.6995 | 0.8853 | 0.8287 |
| +SFT | ViPO-Image-1M-Open | 0.6863 | 0.6070 | 0.5978 | 0.5599 | 0.5995 | 0.8372 | 0.8051 |
| +SFT & Poly-DPO | ViPO-Image-1M-Open | 0.7324 | 0.7013 | 0.6798 | 0.6370 | 0.6791 | 0.8944 | 0.8189 |
| FLUX.1 [dev] | - | 0.6532 | 0.5273 | 0.4491 | 0.4312 | 0.4878 | 0.6727 | 0.7265 |
| +SFT | ViPO-Image-1M | 0.3530 | 0.2462 | 0.1962 | 0.1459 | 0.2126 | 0.4623 | 0.7303 |
| +SFT & Poly-DPO | ViPO-Image-1M | 0.7733 | 0.7203 | 0.6893 | 0.6169 | 0.6859 | 0.8489 | 0.7939 |
| +SFT | ViPO-Image-1M-Open | 0.5551 | 0.5164 | 0.4271 | 0.3715 | 0.4491 | 0.6787 | 0.7499 |
| +SFT & Poly-DPO | ViPO-Image-1M-Open | 0.6327 | 0.5603 | 0.5139 | 0.4231 | 0.5134 | 0.7427 | 0.7656 |

Table 16: Wan2.1-T2V-1.3B Experiments on VBench-2.0 when trained with different datasets.

| Models | Human Identity | Material | Thermotics | Dynamic Spatial Rel. | Dynamic Attribute | Motion Order Und. | Human Interaction | Camera Motion | Motion Rationality |
|---|---|---|---|---|---|---|---|---|---|
| Wan2.1 | 62.18 | 69.75 | 72.26 | 24.64 | 53.48 | 35.35 | 74.00 | 31.79 | 43.68 |
| + ViPO-Video-300K | 67.99 | 71.57 | 68.53 | 33.82 | 57.00 | 38.62 | 78.00 | 32.49 | 47.70 |
| + ViPO-Video-300K-Open | 68.42 | 70.32 | 70.46 | 31.26 | 54.34 | 39.19 | 78.00 | 31.87 | 46.88 |

# F    DISCUSSION, LIMITATION AND FUTURE WORK.

**Discussion.**    Our work presents a dual contribution to scaling visual preference optimization: the Poly-DPO algorithm and the high-quality ViPO dataset. The most significant finding is the symbiotic relationship between algorithmic design and data quality. Our experiments demonstrate that while a robust algorithm like Poly-DPO is critical for extracting meaningful signals from noisy, real-world datasets such as Pick-a-Pic V2, the need for such sophisticated algorithmic adjustments diminishes as data quality improves. The convergence of the optimal Poly-DPO hyperparameter $\alpha$ to zero when training on our ViPO dataset serves as a powerful empirical validation of ViPO's quality and balance.

This suggests that the hyperparameter $\alpha$ can itself serve as a valuable diagnostic tool for assessing preference dataset characteristics. A large positive optimal $\alpha$ may indicate significant noise or conflicting preference signals, whereas a negative optimal $\alpha$ could suggest the dataset is dominated by trivially simple patterns leading to model overconfidence. An optimal $\alpha$ near zero, as observed with ViPO, indicates a well-balanced and reliable dataset where standard optimization is sufficient.

Furthermore, our construction of the ViPO dataset highlights a scalable paradigm for future data curation efforts. By leveraging a suite of state-of-the-art generative models and a panel of powerful Vision Language Models (VLMs) for automated filtering, generation, and labeling, we demonstrate a pipeline that largely bypasses the immense cost and scalability issues of collecting human preferences directly. This AI-driven approach is fundamental to achieving preference optimization "at scale."

**Limitation.**    Despite the promising results, our work has several limitations. First, the preference labels in the ViPO dataset are generated exclusively by AI models (VLMs). While we used multiple state-of-the-art VLMs to ensure robustness and consistency, these AI-generated labels are a proxy for, not a direct measurement of, true human preferences. We did not conduct a large-scale study to measure the correlation between our VLM-assigned labels and those from human annotators, and the inherent biases of the judge VLMs may be encoded in our dataset.

Second, while Poly-DPO's effectiveness is demonstrated across datasets with different characteristics, the optimal value for the hyperparameter $\alpha$ was determined via grid search. This process can be computationally intensive, and the ideal $\alpha$ may depend on factors beyond data noise, such as the base model architecture or the specific domain of the content. A more automated or dynamic method for setting $\alpha$ would improve the method's practicality.

Finally, the creation of the ViPO dataset itself required significant computational resources, involving generation from over a dozen state-of-the-art models. While our work helps democratize the *use* of high-quality preference data through its public release, the initial *construction* of such datasets remains a costly endeavor, potentially limiting the ability of smaller research groups to create similar resources for new domains.

**Future Work.** Based on our findings and limitations, we propose several avenues for future research. A critical next step is to conduct a large-scale human validation study of the ViPO dataset and explore more robust pseudo-labeling with better reward models Chen et al. (2023) and better generative models like Seedance-1.5-Pro Seedance et al. (2025). Comparing the VLM-generated labels against human judgments would not only quantify the quality of ViPO but also provide valuable insights into developing next-generation judge VLMs that are even better aligned with human values.

Another promising direction is the automation of the $\alpha$ hyperparameter in Poly-DPO. Future work could explore methods to make $\alpha$ a learnable parameter that is dynamically adjusted during training based on batch statistics or the model's evolving confidence distribution. This would create a truly self-adaptive preference optimization algorithm.

The categorized structure of the ViPO dataset opens up possibilities for more fine-grained and controllable preference optimization. Future research could investigate methods for explicitly modeling the trade-offs between different quality dimensions (e.g., prioritizing "Text Rendering" over "Aesthetics"), potentially leading to more personalized and instruction-guided visual generation. Lastly, we believe the AI-driven curation pipeline itself can be extended, both to new modalities like 3D and audio, and into an iterative, self-improving loop where models trained on ViPO are used to generate new data that, after being filtered by judge VLMs, is used to further refine the dataset.

# G THE USE OF LARGE LANGUAGE MODELS (LLMs)

All technical content, dataset design, experimental results, and analyses presented in this paper were produced by the authors. Large Language Models (LLMs), such as GPT and Gemini, served only as a tool for language polishing and enhancing readability; they were not used to generate any of the core ideas, data, or experimental results.

## ETHICS STATEMENT

This work develops preference optimization methods and datasets for visual generation models. All experiments were conducted using publicly available models and datasets, with newly generated synthetic data created from text prompts or publicly available image datasets. Our ViPO dataset construction involved AI-generated content from state-of-the-art models (FLUX, Qwen-Image for text-to-image; WanVideo for image-to-video using LAION-Aesthetics images). While we use publicly available datasets that may contain human images, we follow established practices for responsible use of such data. We recognize that visual generation models can potentially be misused for creating misleading or harmful content. To mitigate these risks, we emphasize responsible use guidelines, transparent documentation of our methods, and acknowledge that generated content should be clearly labeled as AI-created. While our work aims to improve generation quality and alignment with human preferences, we encourage ongoing research into detection methods and ethical deployment practices for generative AI systems.

## REPRODUCIBILITY STATEMENT

We are committed to ensuring the reproducibility of our research. A comprehensive description of our dataset construction, including the entire collection and processing pipeline for our proposed ViPO datasets, is provided in Section B. All implementation details, including models, training hyperparameters for each experiment, and the evaluation setup, are thoroughly documented in Section D. We believe these resources provide all the necessary components for the community to reproduce our results and build upon our work.

