# OpenReview forum: "ViPO: Visual Preference Optimization at Scale"
_ICLR.cc/2026/Conference — ICLR 2026 Poster_

### Official Review · Reviewer_kHDq · 2025-10-16

**Soundness:** 4
**Presentation:** 4
**Contribution:** 4
**Rating:** 8
**Confidence:** 5

**Summary:**

This paper introduces a two-pronged approach to advance the scaling of visual preference optimization. The first contribution is Poly-DPO, a novel algorithm that modifies the standard DPO loss with an adaptive polynomial term, enabling more robust learning from datasets with noisy or conflicting preference labels. The second contribution is ViPO, a new large-scale, high-quality dataset of 1 million image pairs and 300,000 video pairs, designed to provide a more reliable and balanced preference signal than existing resources. A key finding is the synergy between these contributions: Poly-DPO significantly outperforms baselines on noisy datasets, but gracefully converges to standard DPO on the high-quality ViPO dataset.

**Strengths:**

This paper presents a significant and compelling contribution to the field of visual preference optimization, marked by its dual focus on both algorithmic innovation and high-quality data curation. My evaluation of this work is highly positive, and its primary strengths are as follows:

*   **Elegant and Effective Algorithm Design:** The proposed **Poly-DPO** is a standout contribution. It is exceptionally well-motivated, directly addressing the critical and practical challenge of learning from datasets with noisy and conflicting preference signals. The design is both simple and powerful—by augmenting the standard DPO loss with a single, intuitive polynomial term, it adaptively re-weights samples based on model confidence. This elegant modification requires minimal code changes yet yields substantial performance gains on noisy datasets, demonstrating a high degree of effectiveness and practical utility.

*   **A Landmark Dataset Contribution:** The introduction of the **ViPO** dataset is a massive contribution to the community. Beyond its impressive scale (1M image and 300K video pairs), its true strength lies in its thoughtful, categorical construction. By organizing preferences into distinct dimensions such as Aesthetics, Text-Image Alignment, and Composition, the authors ensure the dataset is balanced, reliable, and capable of driving comprehensive improvements across a wide range of model capabilities. The data generation strategies are both clever and pragmatic—for instance, creating image-text alignment pairs through prompt perturbation and using OCR to generate robust preference labels for text rendering. This meticulous approach sets a new standard for preference dataset quality.

*   **Comprehensive and Rigorous Experimental Validation:** The paper's claims are substantiated by an extensive and exceptionally well-designed set of experiments.
    *   The **ablation studies** (particularly Figure 4) are cleverly constructed. They not only provide convincing evidence that the "noisy preference" problem is a real and significant obstacle in existing datasets but also clearly demonstrate Poly-DPO's ability to mitigate this issue effectively.
    *   The experiments systematically validate each component of the work. The authors wisely use public benchmarks like the Pick-a-Pic dataset and standard models like SD1.5 to transparently demonstrate the standalone effectiveness of the Poly-DPO algorithm. Subsequently, they showcase the powerful synergy of their algorithm and new dataset, demonstrating state-of-the-art results and significant improvements across a diverse array of modern generative models, including SDXL, FLUX.1, and Wan2.1. This thorough validation leaves little doubt about the efficacy of the proposed methods.

**Weaknesses:**

Despite the paper's numerous strengths and high-quality contributions, there are a few areas that could be further clarified or represent inherent limitations of the chosen approach. These points are offered to help refine what is already an excellent piece of work:

*   **Reliance on AI-Generated Labels for the ViPO Dataset:** The primary limitation is that the ViPO dataset's preference labels are generated exclusively by AI models (e.g., VLMs). While this AI-driven pipeline is a key strength for achieving unprecedented scale, it introduces a fundamental trade-off.

*   **Practicality of Hyperparameter Tuning for Poly-DPO:** While the Poly-DPO algorithm is simple in its formulation, its effectiveness on a new dataset relies on finding an optimal value for the hyperparameter `α` through a grid search. This process can be computationally intensive and presents a practical barrier to easy adoption. The authors correctly identify the automation of `α` as a direction for future work.

*   **Minor Typos and Notational Inconsistencies:** The paper contains several minor but noticeable typographical errors that should be corrected for the final camera-ready version. These do not detract from the core scientific contributions but do affect the overall polish and clarity. For instance:
    *   There is an inconsistency in the naming of the dataset, appearing as both "**ViPO**" and "**VIPO**" in the abstract and elsewhere.
    *   In the introduction, "SD1.5" is incorrectly written as "**SD15**" (line 86).
    A thorough proofread to catch these small errors would improve the professionalism of the final manuscript.

**Questions:**

1. The paper would be even more impactful if it included a small-scale human validation study to measure the correlation between the VLM-generated labels and actual human judgments. By the way, one of related reference papers can be cited (X-IQE: eXplainable Image Quality Evaluation for Text-to-Image Generation with Visual Large Language Models).
2. A discussion on how dataset statistics (like the conflict rate mentioned for Pick-a-Pic) could inform the initial search range for `α` would significantly enhance the method's insight for other researchers.
3. The categorical construction of the ViPO dataset is a major strength. Have you performed any analysis on the relative importance of these different categories (Aesthetics, Composition, etc.)? For instance, do you find that training on certain categories, like Text-Image Alignment, contributes disproportionately to the overall improvements seen on general-purpose benchmarks like GenEval?

---

> ### Author Response · Authors · 2025-11-21
>
> We are deeply grateful for your highly positive evaluation and for recognizing our work as a "significant and compelling contribution" with "elegant and effective algorithm design." Your insightful suggestions have significantly strengthened our manuscript. We have revised the paper (changes marked in blue) and provide detailed responses below.
>
> ---
>
> **Response to W1 & Q1: Human Validation of VLM Labels**
>
> We completely agree. Following your suggestion, we conducted a large-scale human validation study:
> *   **Study Design:** We recruited 18 human annotators to re-evaluate 4,378 pairs from our ViPO dataset (covering both images and videos).
> *   **Results (New Figure 5 & 8):** The results are highly encouraging. Our VLM pipeline achieves an 81.2% agreement rate with the human consensus (majority vote). Remarkably, this is higher than the average agreement of individual human annotators with the consensus (74.7%).
> *   **Conclusion:** This confirms that our VLM-driven pipeline effectively filters out the noise inherent in subjective human labeling, providing a robust and reliable signal for scaling. We have incorporated these results in Section 5.6 in the paper and Section C in the Appendix.
>
> We thank the Reviewer for the insightful suggestion on related paper, we have updated our paper to cite the missing reference,  and may use these methods to further improve the dataset quality in the future.
>
> ---
>
>
> **Response to W2 & Q2: Discussion on Selecting Initial $\alpha$ Range**
>
> > *"What's the recommended value of alpha in poly-loss? How does one tune this hyper-parameter for a new preference dataset?"*
>
> We agree that clear guidance on α is essential. Based on our analysis in **Section 5.2 and Figure 4**, we suggest a simple approach: first, check how consistent the preference labels are across multiple reward models. If consistency is **extremely high** (near 100%), the data is likely over-simplified and α < 0 helps prevent overfitting. If consistency is **low**, the data contains significant noise and α > 0 helps filter out conflicting signals. For **moderate consistency**, which reflects natural subjectivity in human judgment, α ≈ 0 works well.
>
> In practice, tuning is straightforward: performance shows a monotonic trend within each regime, so a simple binary search quickly finds the optimal value. More importantly, for well-balanced datasets like ViPO, performance stays stable across a range around zero (Figure 4c), so fine-grained tuning is often unnecessary.
>
> ---
>
>
> **Response to Q3: Relative Importance of ViPO Categories**
>
> Yes, we performed a detailed ablation study on ViPO sub-datasets, now presented in **Table 11 (Appendix)**.
> *   **Findings:** While training on all datasets yields the best overall performance (validating the comprehensive nature of ViPO), specific categories drive targeted gains. For instance, fine-tuning on the "Text Rendering" subset alone leads to a massive jump in CVTG-2K scores.
> *   **Synergy:** We also observed cross-category benefits; e.g., training on "Alignment" improved "Composition" scores (GenEval), suggesting the model learns generalized structural adherence.
>
> ---
>
>
> **Response to W3: Typos and Inconsistencies**
>
> We have thoroughly proofread the manuscript, unified the dataset naming to "ViPO" throughout, and corrected "SD15" to "SD1.5" (Line 86) along with other minor errors. Thank you for improving the polish of our paper.

---

### Official Review · Reviewer_Cn5U · 2025-10-30

**Soundness:** 2
**Presentation:** 3
**Contribution:** 2
**Rating:** 4
**Confidence:** 5

**Summary:**

This work proposes ViPO, visual preference optimization at scale. It includes two contributions: (a) Poly-DPO which proposes including a polynomial term alongside the standard diffusion loss to dynamically weight preferences in the face of uncertainty in the preference datasets, (b) ViPO dataset, a massive-scale preference dataset with 1M image (1024resolution) pairs across five categories  and 300K video pairs (720P+ resolution) across three categories. Numerical experiments have been done to evaluate the proposed method on preference optimization compared to the baselines.

**Strengths:**

- Large scale preference optimization dataset for image and video generation
- Evaluation on a range of image and video generation models to show the efficacy of the proposed method

**Weaknesses:**

- It is unclear what’s the impact of Polynomial component in Poly-DPO on a decent DPO formulation. Most of the datasets referred in the paper are quite old and would not be used by any modern SOTA T2I models for preference optimization. Besides the alpha becomes yet another hyper-parameter in the DPO formulation which is already quite prone to collapse
- One concerning issue with the VIPO dataset is that SFT with this dataset does not seem to improve models like SD3.5-medium and FLUX.1 dev on DPG bench which is harder benchmark compared to GenEval. This could point to biases in the dataset construction.
- Claims the new insights regarding existing datasets (low resolution images, outdated models, conflicting win/lose pairs) as a contribution, but this has been observed earlier in the literature.

**Questions:**

- New insights regarding existing datasets is not that new. i.e., These datasets also suffer from low resolution (512-768resolution), limited prompt diversity, outdated generation models. Many existing works have demonstrated the same thing. For instance, see RankDPO.
- Why not evaluate SDXL and SD1.5 on DPG-Bench benchmark like in Tab. 5? Why only show the GenEval scores?
- What’s the recommended value of alpha in poly-loss? How does one tune this hyper-parameter for a new preference dataset?
- For the proposed new dataset, since the recommended value of alpha=0 ==> its standard diffusion DPO. Does it mean that Poly-Loss does not help at all? Because the illustrated examples in which alpha!=0, are very old datasets and would no longer be used due to their outdated generation models and low-res images.
- In Tab.5, why does SFT not improve SD3.5-medium and Flux.1 Dev performance? Technically, the ViPO-1M image dataset is better in quality.
- How robust is the proposed Poly-DPO is to model collapse which is observed very frequently with DPO formulation with longer training?

Missing References:
- DSPO — Direct Score Preference Optimization for Diffusion Model Alignment https://openreview.net/forum?id=xyfb9HHvMe
- RankDPO — Scalable Ranked Preference Optimization for Text-to-Image Generation:  https://arxiv.org/abs/2410.18013
- Flow-GRPO— Training Flow Matching Models via Online RL : https://arxiv.org/pdf/2505.05470
- Bridging SFT and DPO for Diffusion Model Alignment with Self-Sampling Preference Optimization https://arxiv.org/abs/2410.05255

---

> ### Author Response · Authors · 2025-11-21
> **Rebuttal (Part-1)**
>
> We thank the reviewer for the constructive feedback. We have updated our paper to address your questions, citing the missing references (RankDPO, DSPO, etc.) and including required experiments in the Appendix.
>
> ---
>
>
> **Response to W1 & Q4: The Use of "Old" Datasets**
>
> > *"Most of the datasets referred in the paper are quite old..."*
>
> > *"...the illustrated examples in which alpha!=0, are very old datasets and would no longer be used..."*
>
> We respectfully clarify the context regarding dataset usage. **Pick-a-Pic v2 is widely recognized as the state-of-the-art open-source preference dataset in the current literature**. Training SD1.5/SDXL on it is the established community standard for fair benchmarking, strictly followed by recent top-tier works including Diffusion-DPO (CVPR 2024) and the reviewer-referenced DSPO (ICLR 2025). We adhere to this protocol to ensure our baselines are rigorous and comparable. Furthermore, the limitations of these aging datasets are precisely the motivation for our work. We constructed ViPO specifically to resolve the "outdated model" and "low resolution" issues, empowering the community with up-to-date signals.
>
> ---
>
> **Response to W1 & Q6: Training Stability of DPO & Poly-DPO**
>
> >*"the DPO formulation which is already quite prone to collapse"*
>
> >*"How robust is the proposed Poly-DPO is to model collapse which is observed very frequently with DPO formulation with longer training?"*
>
> Regarding model collapse, we respectfully clarify that off-policy RL (e.g., Diffusion-DPO) is generally recognized as being more stable than on-policy RL methods (e.g., DDPO), as noted in the Table 1 of original Diffusion-DPO paper. Poly-DPO inherits this stability. In our ablation experiments (please refer to training curves in the new added **Figure 10 of Appendix**), both Standard DPO and Poly-DPO show steady metric improvement until convergence without signs of collapse.
>
> ---
>
>
> **Response to W2 & Q5: SFT Performance on DPG-Bench**
> >  *"SFT with this dataset does not seem to improve models like SD3.5-medium...on DPG-Bench...This could point to biases in the dataset construction."*
>
> > *"In Tab.5 (DPG-Bench), why does SFT not improve SD3.5-medium and Flux.1 Dev performance?"*
>
> We thank the reviewer for this critical observation. To investigate this potential concern, we conducted additional ablation studies in our revision. These new results clarify that the performance plateau stems from optimization trade-offs rather than inherent dataset bias:
> * Sub-datasets Effectiveness (new added **Table 11**): We found that SFT on specific subsets (e.g., "Alignment") significantly boosts DPG-Bench scores (84.24 $\to$ 86.64). This explicitly proves that high-quality alignment signals exist in the data and can be learned effectively.
> * Trade-off Dynamics (new added **Table 12**): Our step-wise analysis reveals that alignment scores actually peak early (at 1000 steps) before the model trades them off to improve Human Quality (which rises continuously). Therefore, the data quality is robust. We selected the final checkpoint to maximize comprehensive performance across all metrics.
>
> Crucially, SFT is merely a warm-up; the state-of-the-art results achieved after applying Poly-DPO (Tables 4-6) demonstrate that our method successfully overcomes these optimization conflicts, achieving superior performance across all metrics simultaneously.
>
> ---
>
>
> **Response to Q2: SD1.5 and SDXL Performance on DPG-Bench**
> > *"Why not evaluate SDXL and SD1.5 on DPG-Bench benchmark like in Tab. 5?"*
>
> We follow the established evaluation protocols from prior works (Diffusion-DPO/Diffusion-KTO/MAPO papers). These methods **do not** use DPG-Bench for SD1.5/SDXL evaluation. To address the potential concern, we have added **Table 10** in Appendix, evaluating SD1.5/SDXL on DPG-Bench. Poly-DPO achieves the highest Overall scores (SD1.5: 67.02, SDXL: 75.67), consistently outperforming Diffusion-DPO and Diffusion-KTO. This further validates our method across all benchmarks. We have also show the results here.
>
>
>
> > **Table 10. Evaluation results on DPG-Bench with the Pick-a-pic V2 training dataset**
> > | Model | Paradigm | Global | Entity | Attribute | Relation | Other | Overall↑ |
> > |-------|----------|--------|--------|-----------|----------|-------|----------|
> > | SD1.5 | Off-Policy | **74.63** | 74.23 | 75.39 | 73.49 | 67.81 | 63.18 |
> > | Diffusion-DPO | Off-Policy | 71.50 | 72.53 | 75.25 | 73.55 | 72.84 | 63.29 |
> > | Diffusion-KTO | Off-Policy | 72.45 | 76.51 | **78.09** | **78.08** | 73.20 | 66.69 |
> > | Poly-DPO | Off-Policy | 73.36 | **78.15** | 76.50 | 75.81 | **73.42** | **67.02** |
> > |-|-|-|-|-|-|-|-|
> > | SDXL | Off-Policy | 83.27 | 82.43 | 80.91 | **86.76** | 80.41 | 74.65 |
> > | Diffusion-DPO | Off-Policy | 83.67 | 83.50 | **81.89** | 81.56 | **81.58** | 75.12 |
> > | MAPO | Off-Policy | 78.22 | 81.31 | 80.65 | 85.35 | 79.85 | 73.80 |
> > | Poly-DPO | Off-Policy | **84.03** | **83.86** | 81.87 | 83.07 | 81.02 | **75.67** |

---

> ### Author Response · Authors · 2025-11-21
> **Rebuttal (Part-2)**
>
> **Response to W3 & Q1: New Insights Regarding Existing Datasets**
>
> > *"Claims the new insights regarding existing datasets..., but this has been observed earlier in the literature."*
>
> > *"New insights regarding existing datasets is not that new.... Many existing works have demonstrated the same thing. For instance, see RankDPO."*
>
>
> We thank the reviewer for this reference and acknowledge that prior works like RankDPO have identified quality issues in existing preference datasets. However, we respectfully argue that **our contribution goes significantly beyond problem identification to provide both algorithmic solutions and large-scale empirical validation:**
>
> 1. While RankDPO and others observed that existing datasets contain noise, they did not propose algorithmic mechanisms to address this during optimization. In contrast, Poly-DPO introduces the first confidence-aware weighting mechanism that dynamically adapts to dataset characteristics, effectively mitigating the impact of conflicting or over-simple preference signals during training.
> 2. Beyond identifying noise, we provide novel insight into why this noise is problematic for scaling. As demonstrated in Figure 1(a), we show that conflicting patterns create a fundamental bottleneck where simply adding more data yields diminishing returns—standard DPO fails to improve beyond a certain point regardless of data scale. This scaling perspective was not explored by prior work.
> 3. We extend the analysis beyond just "noisy" data to include "over-simplified" distributions in Figure 4(b). Poly-DPO addresses this entire spectrum through a single hyperparameter α, providing a unified solution rather than a noise-specific fix.
> 4. We back our insights with unprecedented empirical validation—1M image pairs and 300K video pairs—demonstrating that the identified bottlenecks can be overcome with proper data curation and algorithmic design.
>
> *We have refined our claims in the revised paper (Lines 95-98) to more precisely articulate these distinctions.*
>
> ---
>
> **Response to Q3: How to Select Alpha for New Datasets**
>
> > *"What's the recommended value of alpha in poly-loss? How does one tune this hyper-parameter for a new preference dataset?"*
>
> We agree that clear guidance on α is essential. Based on our analysis in **Section 5.2 and Figure 4**, we suggest a simple approach: first, check how consistent the preference labels are across multiple reward models. If consistency is **extremely high** (near 100%), the data is likely over-simplified and α < 0 helps prevent overfitting. If consistency is **low**, the data contains significant noise and α > 0 helps filter out conflicting signals. For **moderate consistency**, which reflects natural subjectivity in human judgment, α ≈ 0 works well.
>
> In practice, tuning is straightforward: performance shows a monotonic trend within each regime, so a simple binary search quickly finds the optimal value. More importantly, for well-balanced datasets like ViPO, performance stays stable across a range around zero (Figure 4c), so fine-grained tuning is often unnecessary.
>
> ---
>
> **Response to Q4: The Role of Poly-DPO on ViPO Datasets**
>
> > *"For the proposed new dataset, since the recommended value of alpha=0 ==> its standard diffusion DPO. Does it mean that Poly-Loss does not help at all?"*
> **Response to Q4: The Role of Poly-DPO on ViPO Datasets**
>
> > *"For the proposed new dataset, since the recommended value of alpha=0 ==> its standard diffusion DPO. Does it mean that Poly-Loss does not help at all?"*
>
> The convergence to α≈0 on ViPO is a **design validation**, not a limitation—it demonstrates that Poly-DPO is functioning exactly as intended.
>
> Poly-DPO is designed as a **unified framework** where α serves as both an optimization control and a diagnostic signal for dataset quality: α>0 for noisy data (achieving 13.1% gain on Pick-a-Pic V2), α<0 for over-simplified data, and α≈0 for high-quality balanced data like ViPO. As Reviewer kHDq noted, this design is *"elegant and effective... simple and powerful."* The elegance lies in providing **one algorithm that automatically calibrates to diverse data conditions** through a single hyperparameter.
>
> Crucially, Poly-DPO is **never worse than standard DPO**—it provides robustness when needed while gracefully reducing to standard DPO when the data is already high-quality. This makes Poly-DPO a safe, general-purpose choice across any preference dataset.

---

### Official Review · Reviewer_Dmfu · 2025-10-30

**Soundness:** 3
**Presentation:** 3
**Contribution:** 3
**Rating:** 6
**Confidence:** 3

**Summary:**

The paper presents several contributions for diffusion DPO, including large-scale datasets of 1M image pairs and 300K video pairs, and Poly-DPO as an extended algorithm to standard DPO. The new datasets are to address limitations of existing datasets, e.g. low resolution, and conflicting winning patterns. Methods trained on the new datasets got substantial improvements. The Poly-DPO is obtained by taking DPO as binary classification, and Taylor expansion of the standard cross-entropy loss. In the experiments, Poly-DPO outperforms Diffusion-DPO. It can also reveal problems of a dataset for being noisy or over-simple.

**Strengths:**

- The large-scale preference datasets are valuable to the community.
- The Poly-DOP improves the standard Diffusion DPO.
- Experiments are comprehensive and convincing.

**Weaknesses:**

- Poly-DPO is considered as an incremental change to DPO
- Preference pairs are judged by one or more VLMs. It’s not clear about the quality of annotation by aligning with human preference. To improve, authors can conduct cross checks with human raters on a small portion to verify the alignment.
- Minor writing issues
  - Line 075: learning -> learn
  - Line 086: SD15 -> SD1.5

**Questions:**

- How to demonstrate that ViPO doesn't have the problem of conflicting winning patterns?
- How to verify if ViPO is aligned with human preference?

---

> ### Author Response · Authors · 2025-11-21
>
> We explicitly thank the reviewer for recognizing the value of our large-scale datasets, the improvements brought by Poly-DPO, and our comprehensive experiments. We have carefully revised the paper (changes marked in blue) to address your constructive suggestions.
>
> ---
>
> **Response to W1: Novelty of Poly-DPO**
>
> > *"Poly-DPO is considered as an incremental change to DPO"*
>
> We respectfully argue that Poly-DPO is a **fundamental generalization** of the DPO rather than an incremental tweak.
> 1.  **Enabling Scaling on Noisy Data:** As shown in Figure 1(a), standard DPO fails to scale on real-world noisy datasets (e.g., Pick-a-Pic v2) due to conflicting signals. Poly-DPO’s confidence-aware mechanism effectively overcomes this, achieving a 13.1% improvement on HPSv2.1 (Table 2) compared to standard DPO.
> 2.  **A General Framework:** Poly-DPO generalizes standard DPO (which corresponds to the special case $\alpha=0$). As analyzed in Section 5.2 and Figure 4, it provides a unified solution for diverse data distributions: from highly noisy data (requiring $\alpha > 0$) to over-simplified data (requiring $\alpha < 0$). This adaptive capability is essential for scaling preference optimization across varying dataset qualities.
> 3.  **Simplicity as a Merit:** As noted by Reviewer #kHDq, who noted that the method is *"elegant and effective... exceptionally well-motivated... simple and powerful."* We believe that achieving robust scaling with minimal code changes and computational overhead is a significant contribution to the community.
>
> ---
>
> **Response to W2 & Q2: Human Study to Verify Preference Alignment**
>
> > *"...authors can conduct cross checks with human raters on a small portion to verify the alignment."*
>
> Following your insightful suggestion, we conducted a large-scale human evaluation to rigorously verify our VLM-based annotations.
> *   **Setup:** We recruited 18 human annotators to evaluate 4,378 samples from ViPO, covering both image and video datasets (details in Section 5.6 in the paper and  Section C in the Appendix).
> *   **Results (New Figure 5 & 8):**
>     *   Our VLM pipeline achieves a high agreement rate (81.2%) with the majority vote of human annotators (consensus).
>     *   Notably, the VLM pipeline demonstrates higher consistency with the consensus than the average individual human rater (74.7%). For images specifically, VLMs achieve 84.0% accuracy compared to the human average of (74.9%).
> *   **Conclusion:** This study confirms that our VLM-annotated labels are reliable and well-aligned with human preferences, effectively validating the quality of the ViPO dataset.
>
> *We have added these results to Section 5.6 in the paper and Section C in the Appendix.*
>
> ---
>
> **Response to W3: Minor Writing Issues**
>
> We have corrected these typos (Line 075 and Line 086) and proofread the manuscript to fix other minor errors.
>
> *We have corrected these minor issues in the revised manuscript.*
>
> ---
>
> **Response to Q1: Conflicting Patterns in ViPO**
>
> We acknowledge that some degree of conflicting patterns is inherent in visual preference tasks due to subjectivity—even human annotators often disagree. However, we demonstrate that ViPO minimizes harmful systematic conflicts (where a winner in one aspect consistently fails in another) through two evidences:
> 1.  **Robustness via Consensus:** As detailed above, our VLM annotations align better with the human majority consensus (81.2%) than individual human raters do (74.7%). This indicates our pipeline effectively filters out the noise and instability often found in individual judgments.
> 2.  **Empirical Validation ($\alpha \approx 0$):** As shown in Figure 4(c), the optimal Poly-DPO hyperparameter converges to $\alpha \approx 0$ (Standard DPO) on ViPO. This implies that the dataset is sufficiently balanced and reliable for standard optimization, unlike Pick-a-Pic v2 which requires a large $\alpha$ to correct for significant noise and conflicts.

---

### Author Response · Authors · 2025-11-21
**General Rebuttal**

We sincerely thank all reviewers for their thoughtful and constructive feedback. **We have thoroughly revised our paper to address the concerns raised. All major changes are highlighted in BLUE text.**

**Summary of Contributions**. In this work, we identify that conflicting preference patterns in existing datasets fundamentally hinder the scaling of visual preference optimization. To address this, we make three key contributions:
1.  **Poly-DPO:** A novel generalization of the DPO framework that introduces a confidence-aware mechanism, enabling robust scaling across diverse data distributions—from highly noisy to over-simplified patterns.
2.  **ViPO Dataset:** To break the data bottleneck, we construct a massive-scale, high-quality dataset comprising 1M image pairs (1024px) and 300K video pairs(720p+), meticulously categorized to ensure balanced and reliable signals.
3.  **Extensive Validation:** We demonstrate that our approach achieves state-of-the-art results across multiple benchmarks and models (SD1.5, SDXL, FLUX, Wan2.1), validating the synergy between algorithmic adaptability and data quality.

**Summary of Revisions**:
1. **Refined Contribution (Intro, Lines 95-98):** We have sharpened the articulation of our "New Insights," explicitly distinguishing our findings on scaling bottlenecks from prior observations.

2. **Human Evaluation (Section 5.6, Lines 475-529 & Appendix C, Lines 1026-1079)**: We added a comprehensive human study (18 raters, 4,378 samples). Results show our VLM pipeline achieves 81.2% agreement with the consensus, surpassing the average human rater (74.7%).

3. **DPG-Bench on SD1.5/SDXL (Appendix C, Lines 1080-1096 & Table 10)**: We added DPG-Bench evaluations for SD1.5 and SDXL. Poly-DPO achieves the highest Overall scores (67.02 and 75.67, respectively).

4. **SFT Ablation Studies (Appendix C, Lines 1126-1133 & Table 12)**: We provided an analysis of SD3.5-Medium at different SFT steps on DPG-Bench, demonstrating the validity of our dataset.

5. **Training Stability (Appendix C, Lines 1158-1166 & Figure 10)**: We included training curves confirming that Poly-DPO exhibits stable convergence with steadily increasing reward scores and no signs of model collapse.

**Overview of Specific Responses**

*   **To Reviewer `Dmfu`:** We articulated why Poly-DPO represents a crucial generalization rather than an incremental change (W1); conduct a comprehensive user study confirming our AI-annotations are robust and aligned with human judgment (W2, Q1, Q2); and corrected typos (W3).
*   **To Reviewer `Cn5U`:** We clarified the mechanism of Poly-DPO across different distributions and the role of the $\alpha$ hyperparameter (W1, Q3, Q4); justified the use of Pick-a-Pic v2 as the standard benchmark for fair comparison (W1, Q4); provided evidence of training stability for both DPO and Poly-DPO (W1, Q6); presented additional results showing SFT improvements on DPG-Bench** (W2, Q2, Q5); and sharpened the positioning of our contributions (W3, Q1).
*   **To Reviewer `kHDq`:** We conducted a comprehensive user study to verify label robustness (W1); discussed the practicality of tuning $\alpha$ based on dataset statistics (W2, Q2); analyzed the impact of training on specific sub-datasets; and addressed minor issues regarding typos and missing references (W3, Q1).

We once again express our heartfelt gratitude to all reviewers for their valuable time and insights. We hope our responses satisfactorily address all concerns, and we remain open to further discussion.

---

### Author Response · Authors · 2025-12-02
**Rebuttal Summary for ACs and PCs (Part-1)**

Dear ACs and PCs,

We sincerely thank you for your time and effort in handling our submission.

We have **comprehensively addressed all reviewer concerns** through extensive new experiments, systematic analyses, and thorough clarifications. Our revised paper includes a large-scale human evaluation study (18 raters, 4,378 samples), additional benchmark evaluations (DPG-Bench for SD1.5/SDXL), training stability analysis, and refined presentation of our contributions. All major revisions are highlighted in **blue text** in the updated manuscript.

---

## Summary of Contributions

1. **Poly-DPO Algorithm:** A novel generalization of Diffusion-DPO that introduces a confidence-aware mechanism with **only one hyperparameter and two lines of code**, enabling robust training across diverse data distributions from noisy to over-simplified preference patterns.
2. **ViPO Dataset:** To our knowledge, **ViPO is the first large-scale, high-resolution preference dataset for both images (1M pairs, 1024px) and videos (300K pairs, 720p+)**, filling a critical data gap where existing datasets suffer from low resolution, random collection strategies, and outdated generative models (Table 1).
3. **Extensive Validation:** State-of-the-art results across multiple benchmarks and models (SD1.5, SDXL, FLUX, Wan2.1). Notably, Poly-DPO achieves **13.1% improvement** over Diffusion-DPO on HPSv2.1, and our fine-tuned SD3.5-Medium/FLUX models **surpass commercial systems like GPT-Image 1** on DPG-Bench.

---

## Reviewer Feedback Summary

We are grateful that reviewers recognized our significant contributions:

- **Dataset Value** (All Reviewers): Reviewer #Dmfu stated *"The large-scale preference datasets are valuable to the community."* Reviewer #kHDq called it *"a massive contribution to the community"* and *"a landmark dataset contribution."*
- **Algorithm Design** (Reviewers #Dmfu, #kHDq): Reviewer #kHDq mentioned Poly-DPO as *"elegant and effective... exceptionally well-motivated... simple and powerful,"* noting it *"requires minimal code changes yet yields substantial performance gains."* Reviewer #Dmfu praised Poly-DPO as *"Poly-DPO improves the standard Diffusion DPO".*
- **Experimental Validation** (All Reviewers): Reviewer #Dmfu acknowledged that *"experiments are comprehensive and convincing."* Reviewer #Cn5U described our validation as *"Evaluation on a range of image and video generation models ... show the efficacy of the proposed method."* Reviewer #kHDq described our validation as *"comprehensive and rigorous... exceptionally well-designed."*

---

## Clarification on Factual Errors in Reviewer #Cn5U's Assessment

We respectfully note several factual errors in Reviewer #Cn5U's assessment:

### Error 1: The Usage of "Old" Datasets

> *"Most of the datasets referred in the paper are quite old and would no longer be used..."*

This characterization is incorrect. **Pick-a-Pic V2 is widely recognized as the state-of-the-art open-source preference dataset** in the current literature. Training on Pick-a-Pic V2 is the established community standard for fair benchmarking, strictly followed by recent top-tier works including Diffusion-DPO (CVPR 2024) and the reviewer-mentioned DSPO (ICLR 2025).

We adhere to this protocol to ensure rigorous and comparable baselines. Furthermore, the limitations of these existing datasets are precisely the motivation for our work: we constructed ViPO specifically to address the "outdated model" and "low resolution" issues that plague current benchmarks.

*Please refer to Response to #Cn5U W1 & Q4 for details.*

### Error 2: DPO is Prone to Collapse

> *"the DPO formulation which is already quite prone to collapse"*
> *"model collapse which is observed very frequently with DPO formulation"*

This claim conflates different RL paradigms and contradicts established literature. As explicitly documented in **Table 1 of the original Diffusion-DPO paper**, off-policy methods like Diffusion-DPO are generally recognized as being **more stable** than on-policy RL methods (e.g., DDPO, DPOK). The instability and collapse issues are characteristic of on-policy approaches that require iterative sampling, not off-policy DPO. Poly-DPO inherits this stability. Our new added **Figure 10** empirically confirms this: both standard DPO and Poly-DPO exhibit steady metric improvement until convergence with no signs of collapse.

*Please refer to Response to #Cn5U W1 & Q6, and Appendix C Figure 10 (Lines 1158-1166) for details.*

---

> ### Author Response · Authors · 2025-12-02
> **Rebuttal Summary for ACs and PCs (Part-2)**
>
> ## Addressed Concerns
>
> ---
>
> ### 1. Novelty of Poly-DPO
>
> **Addressing Reviewer #Dmfu (W1)**
>
> We clarified that Poly-DPO is a **fundamental generalization** rather than an incremental tweak:
>
> - **Enabling Preference Scaling:** On Pick-a-Pic V2 (the largest open-source preference dataset), standard DPO fails to scale as performance saturates early (Figure 1a). In contrast, Poly-DPO **consistently scales and outperforms DPO across all data scales**, achieving 13.1% improvement on HPSv2.1 (Table 2).
> - **Unified Framework:** Poly-DPO adapts to diverse data distributions (noisy, over-simplified, or high-quality) through a single hyperparameter $\alpha$, with standard DPO as the special case when $\alpha = 0$.
> - **Simplicity is Novelty, Not Limitation:** Poly-DPO introduces only one hyperparameter and two lines of code, yet enables scalable preference optimization across diverse data distributions. We believe such simplicity and effectiveness represent meaningful novelty rather than incremental change. Reviewer #kHDq (confidence: 5) explicitly endorsed this view, praising Poly-DPO as *"elegant and effective... exceptionally well-motivated... simple and powerful."*
>
> Beyond algorithmic innovation, we also contribute ViPO, a large-scale dataset (1M images + 300K videos) recognized by Reviewer #kHDq as *"a landmark dataset contribution."*
>
> *Please refer to Response to #Dmfu W1 in our Rebuttal, and Section 5.2, Figure 1(a), Table 2 in the revised paper for details.*
>
> ---
>
> ### 2. Human Evaluation
>
> **Addressing Reviewer #Dmfu (W2, Q1, Q2) and #kHDq (W1, Q1)**
>
> Following reviewer suggestions, we conducted a **comprehensive and large-scale human evaluation study**:
>
> - **Setup:** 18 human annotators evaluated 4,378 samples from ViPO (both images and videos).
> - **Results:** Our VLM pipeline achieves **81.2% agreement** with human consensus, surpassing the average individual human rater (74.7%). For images specifically, VLMs achieve 84.0% vs. human average of 74.9%.
> - **Conclusion:** This confirms our VLM-annotated labels are reliable and well-aligned with human preferences.
>
> *Please refer to Response to #Dmfu W2 & Q2 and Response to #kHDq W1 & Q1 in our Rebuttal, and Section 5.6 (Lines 475-529), Appendix C (Lines 1026-1079), Figure 5 & 8 in the revised paper for details.*
>
> ---
>
>
> ### 3. Claims on Dataset Insight
>
> **Addressing Reviewer #Cn5U (W3, Q1)**
>
> We clarified that our contribution goes beyond problem identification:
>
> - **From Observation to Solution:** Prior works like RankDPO identified dataset issues but did not propose algorithmic solutions. Poly-DPO is the first confidence-aware mechanism that directly addresses conflicting patterns during optimization.
> - **Scaling Bottleneck Analysis:** We demonstrate that conflicting patterns are the fundamental bottleneck preventing preference scaling (Figure 1a), showing that simply adding data yields diminishing returns without proper handling.
> - **Large-Scale Validation:** We provide unprecedented empirical validation with 1M image pairs and 300K video pairs.
>
> *Please refer to Response to #Cn5U W3 & Q1 in our Rebuttal, and revised Introduction (Lines 95-98) for details.*
>
> ---
>
>
> ### 4. The Role of $\alpha$ on Different Datasets
>
> **Addressing Reviewer #Cn5U (W1, Q4)**
>
> The convergence to $\alpha \approx 0$ on ViPO is a **design validation**. Poly-DPO is designed as a unified framework where **a single hyperparameter $\alpha$ elegantly adapts to diverse data qualities**: positive $\alpha$ for noisy data (achieving 13.1% gain on Pick-a-Pic V2), negative $\alpha$ for over-simplified data, and $\alpha \approx 0$ for high-quality balanced data. This elegant design means practitioners do not need separate algorithms for different dataset qualities. Crucially, Poly-DPO is never worse than standard DPO, making it a safe, general-purpose choice.
>
> *Please refer to Response to #Cn5U Q4 in our Rebuttal, and Section 5.2, Figure 4 in the revised paper for details.*
>
> ---
>
>
> ### 5. SFT Performance and Training Stability
>
> **Addressing Reviewer #Cn5U (W2, Q5, Q6)**
>
> - **SFT Effectiveness:** Table 11 shows that SFT on specific subsets significantly boosts corresponding metrics (e.g., "Alignment" subset improves DPG-Bench from 84.24 to 86.64). Table 12 reveals that the apparent plateau reflects optimization trade-offs, not data quality issues.
> - **Training Stability:** Figure 10 demonstrates that both DPO and Poly-DPO show steady metric improvement with no signs of model collapse, aligning with the conclusion from the original Diffusion-DPO paper.
>
> *Please refer to Response to #Cn5U W2 & Q5, Q6 in our Rebuttal, and Appendix C Tables 11-12 (Lines 1126-1133), Figure 10 (Lines 1158-1166) in the revised paper for details.*

---

> ### Author Response · Authors · 2025-12-02
> **Rebuttal Summary for ACs and PCs (Part-3)**
>
> ## Addressed Concerns
>
> ---
>
> ### 6. Additional Benchmark Evaluations
>
> **Addressing Reviewer #Cn5U (Q2)**
>
> We note that DPG-Bench is not the established evaluation protocol for SD1.5/SDXL in the current literature. Prior works including Diffusion-DPO, Diffusion-KTO, and MAPO papers do not use DPG-Bench for SD1.5/SDXL evaluation. Nevertheless, to address the reviewer's concern, we added DPG-Bench evaluations for SD1.5 and SDXL (Table 10). Poly-DPO achieves the highest Overall scores (SD1.5: 67.02, SDXL: 75.67), consistently outperforming Diffusion-DPO and Diffusion-KTO.
>
> *Please refer to Response to #Cn5U Q2 in our Rebuttal, and Appendix C Table 10 (Lines 1080-1096) in the revised paper for details.*
>
> ---
>
> ### 7. Hyperparameter Guidance
>
> **Addressing Reviewer #Cn5U (Q3) and #kHDq (W2, Q2)**
>
> We suggest a simple approach based on preference consistency across multiple reward models. High consistency indicates over-simplified data ($\alpha < 0$), low consistency indicates noisy data ($\alpha > 0$), and moderate consistency works well with $\alpha \approx 0$. For well-balanced datasets like ViPO, performance stays stable across a range around zero (Figure 4c), so fine-grained tuning is often unnecessary.
>
> *Please refer to Response to #Cn5U Q3 and Response to #kHDq W2 & Q2 in our Rebuttal, and Section 5.2, Figure 4(c) in the revised paper for details.*
>
> ---
>
> ### 8. Sub-dataset Analysis
>
> **Addressing Reviewer #kHDq (Q3)**
>
> We conducted detailed ablation on ViPO sub-datasets (Table 11). While training on all datasets yields the best overall performance, specific categories drive targeted gains (e.g., "Text Rendering" subset significantly boosts CVTG-2K scores). We also observed cross-category benefits, suggesting the model learns generalized capabilities.
>
> *Please refer to Response to #kHDq Q3 in our Rebuttal, and Appendix C Table 11 in the revised paper for details.*

---

> > ### Author Response · Authors · 2025-12-02
> > **Summary for ACs and PCs**
> >
> > This paper presents two main contributions: (1) **Poly-DPO**, a simple and effective generalization of Diffusion-DPO that enables scalable preference optimization across diverse preference patterns, with only one hyperparameter and two lines of code, and (2) **ViPO**, the first large-scale, high-resolution preference dataset for both images (1M pairs) and videos (300K pairs).
> >
> > Reviewers raised concerns regarding: (a) whether Poly-DPO is incremental (#Dmfu W1), (b) alignment of VLM labels with human preferences (#Dmfu W2, #kHDq W1), (c) novelty of dataset insights (#Cn5U W3), (d) the role of $\alpha$ on high-quality datasets (#Cn5U W1, Q4), and (e) training stability (#Cn5U Q6). We have **comprehensively addressed all concerns** through extensive new experiments, including a large-scale human evaluation (18 raters, 4,378 samples), additional benchmark evaluations, and training stability analysis, which are clarified in detail above.

---

### Meta-Review · Area_Chair_fErn · 2026-01-08

**Summary:**

The paper makes two significant contributions: a low-code algorithmic improvement to DPO that handles noisy data distributions and a dataset that addresses the resolution and quality gaps in existing open-source benchmarks. While Reviewer Cn5U initially raised concerns regarding dataset novelty and training stability, the authors provided robust empirical evidence—including training curves and new benchmark evaluations (DPG-Bench)—that successfully refuted claims of model collapse and demonstrated superior performance. Reviewer kHDq (Confidence 5) and Reviewer Dmfu both recognize the work as a high-impact contribution. Based on the rebuttal, the AC expects a positive post-rebuttal consensus and recommends acceptance of the paper.

**Reviewer Concerns:**

Resolved
- Algorithmic Novelty (Reviewers Dmfu, kHDq): Initially seen as a simple Taylor expansion, the authors demonstrated that Poly-DPO is a fundamental generalization that enables preference scaling on noisy data (like Pick-a-Pic V2) where standard DPO saturates.
- Annotation Quality (Reviewer Dmfu): Concerns that VLM-labeled pairs might not align with humans were resolved by a human study involving 4,378 samples. The VLM pipeline showed 81.2% agreement with human consensus.
- Training Stability (Reviewer Cn5U): The concern that DPO is prone to collapse was refuted with new training curves in Appendix C, showing steady metric improvement for both DPO and Poly-DPO without divergence.
- Benchmark Coverage (Reviewer Cn5U): The authors added DPG-Bench evaluations for SD1.5 and SDXL, where Poly-DPO achieved state-of-the-art overall scores, outperforming Diffusion-DPO and KTO.

Outstanding
- Hyperparameter Tuning ($\alpha$): While the authors provided guidance on tuning $\alpha$ based on dataset consistency (positive for noise, negative for simplicity), it remains an additional hyperparameter that practitioners must consider, even if tuning is "coarse-grained."

**Reviewer Scores:**

Reviewer kHDq: This reviewer gave the most enthusiastic feedback, describing the dataset as "landmark" and the Poly-DPO algorithm as "elegant and effective." It is highly likely that the reviewer would keep the original score of 8.

Reviewer Dmfu: Originally concerned about the incremental nature of the algorithm and the VLM-based labeling. The authors' human study (18 raters) showed 81.2% agreement with human consensus, effectively resolving their quality concerns. It is likely that the reviewer would keep their positive rating (maybe keep the score of 6).

Reviewer Cn5U: The most critical reviewer. Their technical objections regarding DPO stability and "old" datasets were mostly addressed and thus an increase of the score is expected.

---

### Decision · Program_Chairs · 2026-01-26

Accept (Poster)